# Fair Minimum Labeling: Efficient Temporal Network Activations for Reachability and Equity

**Lutz Oettershagen**
University of Liverpool
`lutz.oettershagen@liverpool.ac.uk`

**Othon Michail**
University of Liverpool
`othon.michail@liverpool.ac.uk`

## Abstract

Balancing resource efficiency and fairness is critical in networked systems that support modern learning applications. We introduce the *Fair Minimum Labeling* (FML) problem: the task of designing a minimum-cost temporal edge activation plan that ensures each group of nodes in a network has sufficient access to a designated target set, according to specified coverage requirements. FML captures key trade-offs in systems where edge activations incur resource costs and equitable access is essential, such as distributed data collection, update dissemination in edge-cloud systems, and fair service restoration in critical infrastructure. We show that FML is NP-hard and $\Omega(\log |V|)$-hard to approximate, where $V$ is the set of nodes, and we present probabilistic approximation algorithms that match this bound, achieving the best possible guarantee for the activation cost. We demonstrate the practical utility of FML in a fair multi-source data aggregation task for training a shared model. Empirical results show that FML enforces group-level fairness with substantially lower activation cost than baseline heuristics, underscoring its potential for building resource-efficient, equitable temporal reachability in learning-integrated networks.

## 1 Introduction

Resource efficiency and fairness represent dual challenges in designing modern machine learning and networked systems. From large-scale federated learning managing communication budgets across heterogeneous devices [19], to recommender systems scheduling updates and notifications to diverse user groups [18], and distributed sensor networks operating under stringent energy constraints [17], the underlying theme is constant: interactions are dynamic, resources are limited, and equity across participants or data sources is paramount [8, 12, 26]. In such systems, the graph structure captures potential interactions, but the temporal dimension, i.e., when connections are active, governs the actual flow of information, updates, and influence.

Optimizing these temporal interactions solely for aggregate performance or minimal static infrastructure cost often yields suboptimal or unfair outcomes. Simple heuristics might overload communication bottlenecks, drain energy reserves rapidly, or systematically neglect harder-to-reach nodes or groups representing minority populations, specific event types, or distinct data distributions. This can lead to biased data collection pipelines, skewed model training, feedback loops that amplify inequity, and inefficient use of limited resources like energy or bandwidth.

Therefore, principled mechanisms are needed that explicitly co-optimize efficiency and fairness in these dynamic network settings. We consider systems where a central controller has the ability to decide when communication links are active. Our focus is on designing temporal activation patterns over an underlying static graph $G = (V, E)$ that satisfy strong temporal reachability guarantees for distinct node groups, while minimizing the total number of activations. Nodes may represent users, sensors, or agents belonging to specific populations $(V_1, \ldots, V_k)$, and fairness dictates that

39th Conference on Neural Information Processing Systems (NeurIPS 2025).

each group must maintain adequate connectivity over time to designated target nodes. Minimizing the number of edge activations directly translates to reduced energy consumption, communication overhead, and computational load, which is central for sustainable system design [26, 38].

To address this, we introduce the **Fair Minimum Labeling (FML)** problem framework. Conceptually, FML seeks to determine *which* edges in the graph need to be activated and *when*, such that the required number of nodes from *each specified group* can reach the target nodes via paths formed by these time-ordered activations. The primary objective is to achieve this guaranteed fair reachability using the minimum total number of edge activations over time. FML thus provides a formal basis for optimizing resource sparsity while enforcing equitable temporal access.

The introduction of FML as a novel optimization problem is motivated by both practical relevance and theoretical gaps. First, it applies to a wide range of systems where temporal resource allocation directly affects group equity. In distributed data gathering for learning applications, such as those involving environmental sensors, mobile agents, or geographically diverse data streams [31, 43, 42], FML enables minimal-cost activation plans that ensure all groups are fairly represented in the collected data. In edge-cloud systems, it helps decide when communication links should be active so that all device groups can receive timely model updates under bandwidth constraints. Similarly, in infrastructure restoration, FML can guide activation sequences that ensure fair reactivation of services. Second, FML fills a critical gap in the literature. Existing fairness-aware graph methods, such as fair clustering [24], node embeddings [37], and influence maximization [40, 44], are primarily designed for static settings and do not model temporal interactions or activation cost. Fairness has been studied in wireless systems [17], and temporal connectivity has been explored in dynamic networks [22, 27], but fairness constraints remain absent in algorithmic temporal graph design. FML addresses this gap by explicitly minimizing temporal edge activations, treated as a proxy for resource cost, while enforcing group-level temporal reachability. In contrast to decentralized approaches such as classical federated learning [19], where participation is uncoordinated, FML assumes centralized control over link activations, allowing the enforcement of fairness guarantees over time. This design objective distinguishes FML from prior work focused on static structures or continuous flows.

**Our main contributions can be summarized as follows:**

- We introduce the *Fair Minimum Labeling (FML)* problem, a new framework for designing temporally sparse interaction patterns that guarantee group-wise temporal reachability under global resource constraints. We show that FML generalizes classical temporal connectivity problems and prove that it is NP-hard and $\Omega(\log |V|)$-hard to approximate.

- We develop a probabilistic approximation framework based on tree embeddings, yielding two algorithms: (i) an $\mathcal{O}(\log |V|)$-approximation for activation cost, and (ii) a highly efficient bicriteria approximation that guarantees $\mathcal{O}(\log |V|)$ expected cost and a fairness violation factor of $(1 + \varepsilon)^H$, where $H \in \mathcal{O}(\log |V|)$ denotes the height of the embedding tree. Our algorithms address the case of two distinct groups temporally reaching a single terminal.

- We empirically evaluate our algorithms on multi-source learning tasks. The results show that FML-based methods reliably enforce fairness with significantly lower activation cost than fairness-agnostic baselines, while remaining computationally efficient.

The omitted proofs are provided in Appendix A.

## 2 Preliminaries

We denote the set of positive integers (natural numbers without zero) by $\mathbb{N} = \{1, 2, 3, \ldots\}$. For $\ell \in \mathbb{N}$, we use $[\ell]$ to denote the set $\{1, 2, \ldots, \ell\}$. The powerset of a set $X$ is denoted by $\mathcal{P}(X)$.

A *static graph* is an ordered pair $G = (V, E)$, where $V$ is a finite set of *nodes* (or vertices) and $E \subseteq \{\{u, v\} \mid u, v \in V, u \neq v\}$ is a finite set of undirected *edges*. We assume static graphs are simple (no self-loops, no multiple edges between the same pair of nodes). A *colored static graph* $G = (V, E, c)$ with colors $\mathcal{C}$ has an additional function $c : V \to \mathcal{P}(\mathcal{C})$ assigning a (possibly empty) set of colors $c(u)$ to each node $u \in V$ where $\mathcal{C} = [c_{\max}]$ is a set of $c_{\max}$ colors. We use $V_c = \{v \in V \mid c(v) = c\}$. Let $n = |V|$ and $m = |E|$.

A *temporal graph* is an ordered pair $\mathcal{G} = (V, \mathcal{E})$, where $V$ is a finite set of nodes and $\mathcal{E}$ is a finite set of *temporal edges*. Each temporal edge is a pair $e = (\{u, v\}, \tau)$, where $\{u, v\}$ is an undirected pair

of distinct nodes from $V$, and $\tau \in \mathbb{N}$ is the *timestamp* indicating when the interaction between $u$ and $v$ occurs.

Given a static graph $G = (V, E)$, a *temporal labeling* is a function $\lambda : E \to \mathcal{P}(\mathbb{N})$ that assigns a set of timestamps (possibly empty) to each static edge $e \in E$. A labeling $\lambda$ induces a temporal graph $G_\lambda = (V, E_\lambda)$, where the node set is the same as the static graph's, and the set of temporal edges is defined as: $E_\lambda = \{(\{u, v\}, \tau) \mid e = \{u, v\} \in E \text{ and } \tau \in \lambda(e)\}$. The *size* (or *cost*) of a labeling $\lambda$, denoted by $|\lambda|$, is the total number of assigned timestamps, which corresponds to the number of temporal edges in the induced temporal graph, that is, $|\lambda| = |E_\lambda| = \sum_{e \in E} |\lambda(e)|$.

A *temporal path* (or *time-respecting path*) in a temporal graph $\mathcal{G} = (V, \mathcal{E})$ from node $u$ to node $v$ is a sequence of temporal edges $(\{v_0, v_1\}, \tau_1), (\{v_1, v_2\}, \tau_2), \ldots, (\{v_{p-1}, v_p\}, \tau_p)$ from $\mathcal{E}$ such that $v_0 = u$, $v_p = v$, and the timestamps are strictly increasing, i.e., $\tau_1 < \tau_2 < \cdots < \tau_p$. A node $v$ is *temporally reachable* from a node $u$ in $\mathcal{G}$ if such a temporal path exists from $u$ to $v$.

We define the *reachability indicator function* $r(u, v)$ with respect to a given temporal graph $\mathcal{G} = (V, \mathcal{E})$. For any two nodes $u, v \in V$:

$$r(u, v) = \begin{cases} 1 & \text{if } v \text{ is temporally reachable from } u \text{ in } \mathcal{G}, \\ 0 & \text{otherwise.} \end{cases}$$

By convention, we consider every node to be temporally reachable from itself (via a path of length zero), so $r(v, v) = 1$ for all $v \in V$. The specific temporal graph $\mathcal{G}$ (often an induced graph $G_\lambda$) for which reachability is considered will always be clear from the context. Finally, for $U, W \subseteq V$, let $r(U, W) = \sum_{u \in U, v \in W} r(u, v)$ be the sum of pairwise reachability of nodes in $U$ to nodes in $W$.

## 3   Related Work

The Fair Minimum Labeling (FML) problem intersects several research areas, primarily temporal graph algorithms, network optimization with fairness considerations, and resource-efficient AI.

**Temporal Graph Connectivity and Labeling.** A core component of FML is finding a minimum-cost set of temporal edge activations. Mertzios et al. [27] (and its journal version [28]) introduced the study of temporal network design in multi-labeled temporal graphs, where the goal is to optimize some global cost measure of the graph labeling, such as the total number of timestamps, while satisfying specific temporal reachability properties. The *Minimum Labeling (ML)* problem is such a temporal network design problem, seeking the minimum number of labels to make an entire static graph temporally connected. For its directed version on strongly connected digraphs, [27] had provided bounds on the number of labels needed, proving that 2 labels per arc and at most $4(|V| - 1)$ total labels are always sufficient. Using similar constructions on undirected graphs, Akrida et al. [2] proved that $2|V| - 3$ total labels are sufficient, later refined by Klobas et al. [25] to $2|V| - 4$ under certain conditions, who also showed ML is polynomially solvable. While ML aims for global temporal connectivity, FML focuses on targeted reachability for specific node groups to a designated terminal set, under explicit fairness (coverage) constraints for each group, and a cost minimization objective. Klobas et al. [25] also introduced extensions like *Minimum Steiner Labeling (MSL)*, where the goal is to pairwise temporally connect a predefined set of terminal nodes. Our FML generalizes ML and MSL. Other research has explored concepts like temporally-connected spanning subgraphs [3], temporal core decompositions [15, 33], and various notions of temporal paths and reachability [22, 7, 29, 16, 34, 35, 30]. However, these works generally do not incorporate group fairness constraints or the specific cost model of minimizing temporal edge activations as FML does.

**Fairness in Graph Mining and Network Optimization.** There is a significant body of work on incorporating fairness into static graph algorithms and network optimization, driven by the need for equitable outcomes in various applications. This includes, e.g., fair clustering [1, 10, 24], fair node embeddings [37, 6], fair influence maximization [40, 44], and graph learning tasks [9, 11, 32, 45, 36]. While these contributions are crucial for advancing fairness in graph-based AI, they predominantly operate on static graphs and often define fairness differently (e.g., proportionality in output, balanced representation in a static structure). For example, a related problem in static graphs is the *Balanced Connected Subgraph (BCS)* problem [5] which seeks a connected subgraph with balanced representation from different node colors. FML, in contrast, addresses fairness through *temporal path-based accessibility* for predefined groups, directly optimizing the dynamic activation of interactions over time to meet these fairness goals while minimizing resource usage. Moreover,

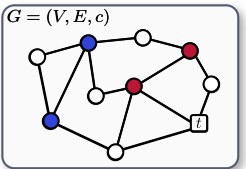
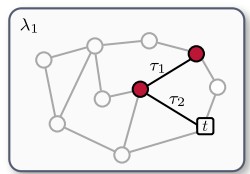
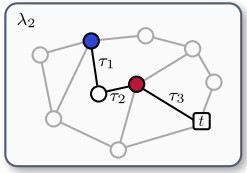

(a) Example of a colored static graph $G$ with one terminal node $t \in V$ shown as square.

(b) Temporal labeling $\lambda_1$ connects $50\%$ of the colored nodes to $t$, but only red nodes.

(c) Temporal labeling $\lambda_2$ connects $50\%$ of the colored nodes in equal proportion to $t$.

Figure 1: A simple toy example showing a non-equitable and an equitable labeling of a small colored graph. The labelings $\lambda_1$ and $\lambda_2$ assign time stamps $\tau_1 < \tau_2 < \tau_3$ to edges. Both connect $50\%$ of the colored nodes, but only $\lambda_2$ connects both colors resulting in higher costs of three. Therefore, under a color requirement of $50\%$ for each color, only $\lambda_2$ is a valid solution for the FML problem.

in networked systems, fairness is a long-standing concern in resource allocation. This includes fair bandwidth sharing in communication networks [21], fair scheduling in wireless systems [17, 41], and equitable service provisioning. In edge-cloud environments, fair scheduling of updates (e.g., for recommender systems [18]) or computation offloading is critical. FML contributes to this line of work by providing a formal model for achieving group-level temporal reachability fairness specifically through the lens of minimizing discrete edge activations, a key proxy for communication cost, energy, or computational overhead, aligning with sustainability principles [26, 38]. Unlike other problems that focus on flow rates or latency targets, FML provides hard guarantees on temporal path existence for specified quotas from diverse groups.

**Connection to Combinatorial Optimization.** FML resembles Set Cover and Steiner Tree [20], with its goal of covering group nodes via minimal-cost temporal activations. Its $\Omega(\log |V|)$ approximation hardness matches Set Cover [14], but FML is more complex due to temporal constraints, labeling-based edge induction, and fairness across groups. Standard algorithms fall short; our approximation algorithms use probabilistic tree embeddings to address FML's structural challenges.

## 4   The fair minimum labeling problem

We now formalize the central optimization problem studied in this work: designing temporally sparse activation patterns over a static graph to guarantee group-level access to designated target nodes under strict resource constraints. In many learning-integrated systems, such as data aggregation, distributed monitoring, or update dissemination, ensuring both efficiency and equitable access over time is essential. Existing frameworks, however, lack mechanisms to enforce fairness through temporal reachability while minimizing total activation effort.

The *Fair Minimum Labeling (FML)* problem addresses this gap by integrating three key aspects: (i) cost-aware edge activation planning, (ii) per-group temporal reachability guarantees, and (iii) flexible, group-specific coverage requirements. The goal is to compute a temporal edge labeling that minimizes total activation cost while satisfying fairness constraints across all groups (see Appendix B for an application example).

**Definition 1** (Fair Minimum Labeling (FML)). *Given a colored static graph $G = (V, E, c)$ with node colors $\mathcal{C}$, a set of terminals $\mathcal{T} \subseteq V$, $\rho \in \mathbb{N}$, and a requirement function $f_c : V \to \mathbb{R}$ for each color $c \in \mathcal{C}$, the goal is to find a temporal labeling $\lambda : E \to \mathcal{P}(\mathbb{N})$ that minimizes the total number of assigned time labels $|\lambda| = \sum_{e \in E} |\lambda(e)|$ such that for each color $c \in \mathcal{C}$, the number of nodes in $V_c$ that can temporally reach at least $\rho$ terminals $t \in \mathcal{T}$ under $\lambda$ is at least $f_c(V_c)$. Reachability is defined with respect to the temporal subgraph induced by $\lambda$.*

Figure 1 shows a simple example. In the corresponding decision version, we are additionally given a budget $k \in \mathbb{N}$ and asked whether there exists a temporal labeling $\lambda$ such that $|\lambda| \le k$ while satisfying the fairness constraints for all groups. We also note that a symmetric variant of the problem can be defined by requiring temporal reachability *from* (rather than *towards*) the terminal set $\mathcal{T}$.

The FML problem generalizes both the Minimum Labeling (ML) and the Minimum Steiner Labeling (MSL) problems (see Observation 1 in the appendix). Since MSL is NP-complete [25], it immediately

follows that FML is NP-complete. Moreover, we show that FML remains NP-hard and hard to approximate within a factor better than $\Omega(\log n)$ even in the special case where the terminal set is a singleton, i.e., $\mathcal{T} = \{t\}$.

**Theorem 1.** *The FML problem with a single terminal is NP-complete, and no $((1 - \epsilon) \log n)$-approximation algorithm exists unless $NP \subseteq DTIME[n^{\mathcal{O}(\log \log n)}]$.*

## 5 Approximation

Given the hardness results in Theorem 1, we now present approximation algorithms that are tight in expectation with respect to cost and solve the FML problem in instances with a single terminal. We focus on the case where each node has colors $B$ (blue), $R$ (red), or no color, i.e., $\mathcal{C} = \{B, R, \emptyset\}$. For convenience, we use the set notation $B \subseteq V$ and $R \subseteq V$ to directly refer to the nodes of each group of colored nodes. Furthermore, we define the group-specific requirement functions as $f_B(B) = \alpha|B|$ and $f_R(R) = \alpha|R|$ for a fixed parameter $\alpha \in \mathbb{R}$ with $0 \leq \alpha \leq 1$, indicating that an equal fraction $\alpha$ of both $B$ and $R$ nodes must reach the terminal. This two-group setting captures a common fairness use case and provides a natural foundation for future extensions to multiple groups or terminals.

Our approach leverages the framework of *probabilistic tree embeddings*, a powerful technique for approximating arbitrary metric spaces with simpler tree metrics. The concept of probabilistic tree embeddings was first introduced by Bartal [4], who showed that any finite metric space can be probabilistically approximated by a distribution over tree metrics, incurring only a logarithmic distortion in expectation. This result was later significantly improved by Fakcharoenphol, Rao, and Talwar (FRT) [13], who established that any $n$-point metric space can be embedded into a distribution over dominating tree metrics with an expected stretch of $\mathcal{O}(\log n)$, and that this bound is tight.

**Definition 2** (Probabilistic Tree Embedding). *Let $(X, d)$ be a finite metric space. A* probabilistic tree embedding *is a randomized mapping from $(X, d)$ into a distribution $\mathcal{D}$ over tree metrics $(X, d_T)$ such that for all $x, y \in X$:*

1. *For every $(X, d_T) \in \mathcal{D}$, $d(x, y) \leq d_T(x, y)$*

2. *The expected stretch is $\mathbb{E}_{d_T \sim \mathcal{D}}[d_T(x, y)] \leq \mathcal{O}(\log n) \cdot d(x, y)$ for all $x, y \in X$.*

---

**Algorithm 1:** Approximation Framework for FML with Single Terminal

**Input:** Graph $G = (V, E)$, $B \subseteq V$, $R \subseteq V$, terminal $t \in V$, $\alpha \in [0, 1]$
**Output:** Temporal graph labeling $\lambda_G$

1 Compute shortest path metric $d_G$ on $G$.
2 Generate a probabilistic tree embedding for $(V, d_G)$ using FRT [13], sample a single weighted tree $T = (V, E_T, w)$ from the distribution $\mathcal{D}$. Let $w(u, v) = d_T(u, v)$.
3 Root the tree $T$ at the terminal $t$.
4 Compute an FML solution $\lambda_T$ for the weighted tree $T$ (rooted at $t$) with groups $R, B$ and parameter $\alpha$, minimizing the weighted cost $C_T = \sum_{(\{u,v\},\tau) \in \lambda_T} w(u, v)$, using either the exact (Section 5.1) or approximate (Section 5.2) algorithm.
5 Project the tree solution $\lambda_T$ back to a solution $\lambda_G$ in the original graph $G$ (Section 5.3).
6 **return** $\lambda_G$

---

Algorithm 1 outlines our general framework for approximating the Fair Minimum Labeling (FML) problem using probabilistic tree embeddings. We begin with the input graph $G = (V, E)$ and the FML parameters $(R, B, t, \alpha)$. We define the standard shortest path metric $d_G$ on $G$, where $d_G(u, v)$ is the minimum number of edges on a path between $u$ and $v$. We then employ the probabilistic tree embedding algorithm of Fakcharoenphol, Rao, and Talwar [13]. In practice, we first sample a single weighted tree $T = (V, E_T, w)$ from the distribution $\mathcal{D}$ and then solve an adapted version of the FML problem on the sampled weighted tree $T$. We use the original node set $V$ with colors $B$ and $R$, terminal $t$, and the parameter $\alpha$. The objective, however, is modified to align with the structure provided by the embedding and the goal of minimizing the final cost in $G$. We aim to find a temporal labeling $\lambda_T$ for the tree $T$ that minimizes the total weighted cost

$$C_T = \sum_{(\{u,v\},\tau) \in \lambda_T} w(u, v) = \sum_{(\{u,v\},\tau) \in \lambda_T} d_T(u, v)$$

subject to the coverage constraints being satisfied with respect to temporal reachability within the tree $T$, requiring that an $\alpha$-fraction of the nodes in each of B and R can temporally reach the terminal $t$.

To achieve this, we use dynamic programming algorithms presented in the following sections:

- An exact algorithm (Section 5.1) finds the minimum weighted cost $C_{T,\text{opt}}$ satisfying the constraints exactly on $T$.
- An approximation algorithm (Section 5.2) finds a labeling $\lambda_T$ with weighted cost $C_T \leq C_{T,\text{opt}}$ that satisfies the constraints approximately, introducing an error factor $\xi \leq (1+\varepsilon)^{H+1}$ on the counts of blue and red nodes, where $H$ is the height of $T$ and $\varepsilon > 0$.

Finally, in Section 5.3, we discuss how the solution obtained in the tree can be projected back to the graph $G$ and $\lambda_G$ can be obtained.

## 5.1 An exact algorithm for FML in weighted trees

We compute labels at the nodes of the tree $T$ in a bottom-up fashion starting from the leafs. The edges of the tree have weights $w(u,v) \geq 1$ corresponding the shortest paths distance between $u, v \in V$ in $G$. Without loss of generality the maximum distance (in terms of edge weights) from any leaf to the root is at most $k$. Furthermore, all leafs are colored blue or red. We use labels of the form $(b, r, c)$ where $b$ and $r$ are the numbers of blue and red nodes in the subtree, and $c$ is the sum of edge weights used to connect the $b + r$ nodes to the node having the label. For example, a blue node gets a label $(1, 0, 0)$ and a red node $(0, 1, 0)$. Let $v$ be an inner node with $\ell$ children $u_1, \ldots, u_\ell$. We construct the following labels based on the labels of the children. Let $\{L_1, \ldots, L_\ell\}$ be the sets of labels at $u_1, \ldots, u_\ell$. We only choose at most one label per child that determines the (possibly empty) subtree to include. Therefore, we construct at most $\prod_{i=1}^{j=\ell}(1 + |L_j|)$ labels at node $v$. At each node we keep at most one label $(b, r, c)$ for each possible pair $(b, r)$ with the minimum value of $c$. Because $r$ and $b$ are at most $n$ there are $\mathcal{O}(n^2)$ possibilities of $(b, r)$ which is also an upper bound of the number of (non-dominated) labels at each node. Algorithm 2 in the appendix shows the pseudocode.

**Theorem 2.** *The FML problem on a tree with a single terminal can be solved in $\mathcal{O}(n^5)$.*

## 5.2 A bicriterial approximation

The running time of $\mathcal{O}(n^5)$ of the exact algorithm for solving the FML problem in trees limits its scalability. The bottleneck is the linear number of labels per node, and the pairwise merging costing $\mathcal{O}(n^4)$ per node. So the idea is to reduce the number of labels per node in a principled way. A classic method to reduce dynamic programming overhead is label sparsification by grid rounding. The key is to group labels with similar $(b, r)$ values into a common bucket and keep only the best, i.e., lowest cost $c$, label in that bucket.

Specifically our approach partitions the integer range $[0, n]$ into *geometric intervals*, or buckets, defined as follows: Bucket $i$ corresponds to the interval $((1 + \varepsilon)^{i-1}, (1 + \varepsilon)^i]$ for $i \geq 1$ until the upper bound $n$ is reached. This results in $\mathcal{O}(\log_{1+\varepsilon}(|B|) \cdot \log_{1+\varepsilon}(|R|)) = O\left(\frac{1}{\varepsilon^2} \log^2 n\right)$ buckets for possible $(b, r)$-bucket pairs. Once the label set $L_v$ is constructed at a node $v$, each label $(b, r, c)$ is mapped to a pair of bucket indices $(\text{BucketIndex}(b), \text{BucketIndex}(r))$, and only the label with the minimum cost $c$ is retained per bucket. All other labels in the same bucket are discarded, thereby reducing the label set to at most $O\left(\frac{1}{\varepsilon^2} \log^2 n\right)$ entries.

**Theorem 3.** *Let $\varepsilon > 0$. Let $T$ be the input tree with root $t$ and height $H$. Let $\xi = (1 + \varepsilon)^{H+1}$. Let $\ell_{\text{opt}} = (b_{\text{opt}}, r_{\text{opt}}, c_{\text{opt}})$ be an optimal exact label at the root $t$ with minimum weighted cost $c_{\text{opt}}$ that satisfies $b_{\text{opt}} \geq \alpha|B|$ and $r_{\text{opt}} \geq \alpha|R|$. Then the geometric bucketing algorithm finds a label $\ell' = (b', r', c')$ in the final set $L_t'$ at the root with*

- *(i) $c' \leq c_{\text{opt}}$,*
- *(ii) $b' \geq b_{\text{opt}}/\xi \geq (\alpha|B|)/\xi$ and $r' \geq r_{\text{opt}}/\xi \geq (\alpha|R|)/\xi$.*

## 5.3 Projecting the solution to the graph G

Let $\lambda_T$ be the temporal labeling obtained by one of these algorithms on the tree $T$. The tree labeling $\lambda_T$ must be translated into a valid temporal labeling $\lambda_G$ for the original graph $G$. This projection

ensures that the reachability achieved in the tree is preserved in the graph. For each temporal tree edge activation $(\{u, v\}, \tau) \in \lambda_T$ we first identify a shortest path $P_{uv}$ between $u$ and $v$ in the original graph $G$. Let this path consist of the sequence of $p = d_G(u, v)$ edges $e_1, e_2, \ldots, e_p$. To mimic the activation of the tree edge $\{u, v\}$ at time $\tau$, we activate the path $P_{uv}$ in $G$ using a sequence of timestamps. The final graph labeling $\lambda_G$ is the union of all temporal edges generated by projecting all $(\{u, v\}, \tau) \in \lambda_T$.

The projection mechanism ensures that if node $w$ is temporally reachable from node $u$ in the tree $T$ using the labeling $\lambda_T$, then $w$ is also temporally reachable from $u$ in the graph $G$ using the labeling $\lambda_G$. This holds because each temporal tree edge in a path is replaced by a corresponding temporal path segment in $G$, and the timestamping ensures correct sequencing and avoids conflicts. Therefore, if $\lambda_T$ satisfies the coverage constraints of Definition 1 (either exactly or approximately with factor $\xi$), the projected solution $\lambda_G$ will also satisfy these constraints either exactly or approximately with the same factor $\xi$ in the original graph $G$. Note that we potentially include additional colored nodes that will be able to reach the terminal $t$, however, this can only improve the approximation.

**Theorem 4.** *Algorithm 1, combined with the appropriate weighted tree algorithm, yields a randomized algorithm for FML with one terminal on general graphs with two colors.*

- *If the exact weighted tree algorithm (Section 5.1) is used, the framework produces a solution $\lambda_G$ that satisfies the fairness coverage constraints exactly, and its expected cost $\mathbb{E}[|\lambda_G|]$ is within an $\mathcal{O}(\log n)$ factor of the optimal cost $k_G^*$. The expected running time is in $\mathcal{O}(n^5)$.*

- *If the bicriteria weighted tree algorithm (Section 5.2) is used, the framework yields a solution $\lambda_G$ with expected cost $\mathbb{E}[|\lambda_G|] \leq \mathcal{O}(\log n) \, k_G^*$ that approximately satisfies the coverage and fairness constraints, with violation bounded by the factor $\xi = (1 + \varepsilon)^{H+1}$, where $H \in \mathcal{O}(\log n)$. This results in a randomized bicriteria $(\mathcal{O}(\log n), \xi)$-approximation.*

  *The expected running time is in $\mathcal{O}(n^2 + n\varepsilon^{-4} \log^4 n)$.*

*Both variants use $\mathcal{O}(n)$ space for the tree embedding and $\mathcal{O}(n)$ (exact) or $\mathcal{O}(n\varepsilon^{-2} \log^2 n)$ (bicriteria) additional space for the DP tables.*

### 5.4 Generalizations

The algorithms and the specific complexity analysis presented in the previous sections are for the two-group (plus no-color) case. However, the framework is generalizable. For $k$ distinct groups, the dynamic programming label becomes $(g_1, g_2, \ldots, g_k, c)$, where $g_i$ is the count of covered nodes from group $i$. The state space for the counts becomes $\mathcal{O}(n^k)$. The pairwise merge operation takes $\mathcal{O}((n^k)^2) = \mathcal{O}(n^{2k})$ time. With $n$ nodes, the total runtime for the exact DP is $\mathcal{O}(n^{2k+1})$.

Our framework can also be extended to support non-uniform activation costs $w(e)$, while preserving the $\mathcal{O}(\log n)$ approximation guarantee, by defining the metric for the FRT embedding using these costs. First, we compute a shortest-path metric $d(u, v)$ where the distance between nodes is defined as the minimum cost of a path between them, using the activation costs $w(e)$ as the edge weights. We then use FRT to embed this weighted metric into a tree $T$. Our DP algorithm then runs on the tree $T$, and because the tree distances being optimized are already low-distortion approximations of the minimum activation costs, the $\mathcal{O}(\log n)$ guarantee on the final, total weighted cost holds directly.

Finally, in dynamic environments (e.g., node arrivals, failures or departures), partial recomputation of our labeling is feasible. The DP can be rerun locally on affected subtrees, and minor changes in the graph structure often preserve the validity of the tree embedding. While full dynamic adaptation is beyond the scope of our current work, the methods lend themselves to such extensions.

## 6 Experiments

In this section, we evaluate the performance of our algorithms. We compare the following methods:

- FMLAPPROX, our approximation algorithm based on Algorithm 1 that uses the exact tree dynamic programming subroutine described in Section 5.1.
- FMLBIAPPROX, our bicriteria approximation algorithm based on Algorithm 1 that uses the approximate tree dynamic programming subroutine described in Section 5.2.

We designed the following greedy baselines for comparison as there are no direct competitors to our approach, where we start with two natural starting points, (i) what happens if you ignore fairness entirely and focus only on cost? (ii) what are the simplest, most direct ways to enforce fairness? Leading to the following baselines:

- GREEDY is a fairness-agnostic baseline that activates shortest paths to the closest colored nodes until $\alpha \cdot (|B| + |R|)$ total colored nodes are covered, irrespective of group balance.

- CLOSEST greedily activates unweighted shortest paths from the terminal to the closest uncovered colored node (regardless of group) until $\alpha$-coverage is achieved for both groups.

- ALTERNATING proceeds similar to Closest, but alternates between activating the closest uncovered node from group $B$ and group $R$.

We implemented all algorithms in Python 3.9 and PyTorch 2.5.1. All experiments ran on a computer cluster. Each experiment ran exclusively on a node with an Intel(R) Xeon(R) Gold 6130 CPU @ 2.10GHz, 96 GB of RAM. We used a time limit of 12 hours. Our source code is available at https://gitlab.com/tgdesign/fml .

## 6.1 Case study: Fair and efficient multi-source data collection

We evaluate our FML framework in a multi-source learning context, where data must be collected from distributed sources under strict resource constraints. Such sources, e.g., sensors, mobile agents, or geographically distributed streams, often differ by group and face limitations like bandwidth or energy. Ensuring fair representation of each group during training is essential to prevent biased models. However, naive approaches, such as connecting all sources, minimizing total cost, or favoring proximity, either waste resources or neglect important groups. Our method computes a single temporal edge labeling that guarantees sufficient group-wise temporal reachability to a central node while minimizing the number of activations. This enables efficient and fair data collection without the need for repeated adaptation or full connectivity, outperforming the baselines.

### 6.1.1 Network topology and data sources

We simulate a networked learning scenario over a random geometric graph $G = (V, E)$ with $|V| = 1024$ nodes and connection radius $r = 0.2$. The designated terminal node $t \in V$ collects data from a subset of data sources $B \cup R \subseteq V \setminus \{t\}$. Each of the data sources belongs to one of two demographic groups $B$ or $R$, assigned based on spatial proximity to the terminal (near vs. far).

Each data source generates synthetic data $(\mathbf{x}, y)$ with $\mathbf{x} \in \mathbb{R}^2$, $y \in 0, 1$ by sampling $\mathbf{x}$ from a mixture of two Gaussians and setting $y$ deterministically:

- Group $B$: $\mathbf{x} \sim \mathcal{N}(\mu, \sigma^2 I)$ with $\sigma = 0.55$, $\mu \in \{(-2, 1), (2, 1)\}$, and $y = \mathbf{1}\{x_1 > 0\}$.
- Group $R$: $\mathbf{x} \sim \mathcal{N}(\mu, \sigma^2 I)$ with $\sigma = 0.55$, $\mu \in \{(1, -2), (1, 2)\}$, and $y = \mathbf{1}\{x_2 < 0\}$.

Thus, the two groups encode orthogonal classification rules, making it impossible to achieve high accuracy on both without training on both.

The algorithms activate edges $A \subseteq E$ to connect data sources to the terminal. For the baselines, the activation cost is defined as the sum of hop-distances from all covered data sources to the terminal. The collected data is used to train a shared MLP classifier. We use a fully connected MLP using an input layer of size $d = 2$, followed by two hidden layers with 32 neurons each and ReLU activations, and a final output layer with a single neuron and sigmoid activation for binary classification. We train the model using the Adam optimizer [23] with learning rate $0.01$, batch size $64$, and weight decay $10^{-2}$ for $\ell_2$ regularization. The loss is binary cross-entropy. Training runs for 100 epochs with early stopping based on validation loss. The model is trained using the combined dataset collected from data sources temporally connected to the terminal. Accuracy is reported separately on held-out test sets for each group. We also report the coverage of nodes in $B$ and $R$ that can temporally reach $t$. We report the mean values and standard deviations over ten independent runs.

The results in Table 1 demonstrate the trade-offs between fairness, cost, and accuracy across different methods. The straightforward GREEDY, while achieving the low activation cost (88.3), completely neglects group $R$ ($\text{Coverage}_R = 0$), leading to severe disparity in model accuracy between groups with

Table 1: Multi-source learning results (mean and standard deviations over ten independent runs).

| Algorithm | Time (s) | Cost | $\text{Coverage}_B$ | $\text{Coverage}_R$ | $\text{Acc}_B$ | $\text{Acc}_R$ |
|---|---|---|---|---|---|---|
| GREEDY | $0.49 \pm 0.08$ | $88.3 \pm 6.25$ | $64.0 \pm 0.00$ | $0.0 \pm 0.00$ | $1.00 \pm 0.00$ | $0.27 \pm 0.17$ |
| CLOSEST | $0.73 \pm 0.13$ | $224.0 \pm 0.00$ | $44.3 \pm 2.31$ | $32.0 \pm 0.00$ | $0.97 \pm 0.04$ | $0.94 \pm 0.08$ |
| ALTERNATING | $0.71 \pm 0.09$ | $212.8 \pm 1.81$ | $34.2 \pm 1.87$ | $32.0 \pm 0.00$ | $0.96 \pm 0.04$ | $0.94 \pm 0.07$ |
| FMLAPPROX | $69.20 \pm 18.11$ | $74.3 \pm 4.62$ | $33.5 \pm 0.53$ | $32.3 \pm 0.48$ | $0.98 \pm 0.03$ | $0.93 \pm 0.08$ |
| FMLBIAPPROX | $8.54 \pm 6.36$ | $73.9 \pm 3.48$ | $33.1 \pm 0.99$ | $31.3 \pm 0.48$ | $0.97 \pm 0.03$ | $0.94 \pm 0.08$ |

$\text{Acc}_B = 1.00$ and $\text{Acc}_R = 0.27$. This highlights the risk of fairness-agnostic optimization. CLOSEST and ALTERNATING heuristics ensure coverage for both groups, but incur significantly higher costs with 224.0 and 212.8, respectively. In contrast, both our FML-based algorithms, FMLAPPROX and FMLBIAPPROX, achieve near-perfect group-level fairness ($\text{Coverage}_B \approx 33$, $\text{Coverage}_R \approx 32$) with substantially lower cost of around 74, and maintain high accuracy for both groups with $\text{Acc}_B \geq 0.97$ and $\text{Acc}_R \geq 0.93$. FMLBIAPPROX performs comparably to the exact method in terms of fairness and accuracy, while being significantly faster ($8.54s$ vs. $69.20s$ on average). These results confirm the effectiveness of FML in achieving resource-efficient and equitable temporal reachability.

We also ran additional experiments where we explicitly placed the terminal at either a central (high-degree) or peripheral (low-degree) node. Our bicriteria algorithm (FMLBIAPPROX) proved highly effective in both scenarios, consistently achieving over 98% of the required coverage.

## 6.2 Running time and approximation quality

We additionally compare the performance of our approximations FMLBIAPPROX and FMLAPPROX. To this end, we generated Barabási–Albert graphs with $n \in \{256, 512, 1024, 2048, 4096\}$ and fixed attachment parameter $\bar{m} = 3$ (i.e., each new node connects to $\bar{m}$ existing nodes). Moreover, we chose $\varepsilon \in \{0.1, 0.01, 0.001\}$. For each parameter combination, we used ten independent runs. Figure 2a reports the average running time and standard deviation across all in the time limit completed runs. For the largest graphs, FMLAPPROX failed to complete five instances for $n = 2048$ and seven for $n = 4096$, while FMLBIAPPROX timed out on three and five runs, respectively. Among the runs that finished, FMLBIAPPROX consistently outperformed FMLAPPROX, achieving up to an order-of-magnitude speedup. Importantly, decreasing $\varepsilon$ led to only modest increases in running time, yet produced significant gains in solution quality. Figure 2b shows the mean ratio of $B$ ($R$, resp.) colored nodes selected by FMLBIAPPROX compared to FMLAPPROX. Smaller values of $\varepsilon$, particularly $\varepsilon = 0.001$, yield ratios close to 1, indicating near-perfect agreement with the exact baseline. These results demonstrate the effectiveness of FMLBIAPPROX in achieving high-quality solutions with substantially reduced running time. Finally, the majority of the running time, more than 99.9% across all settings, is spent on solving the dynamic program on the FRT tree. The time spent on tree embedding is negligible in comparison.

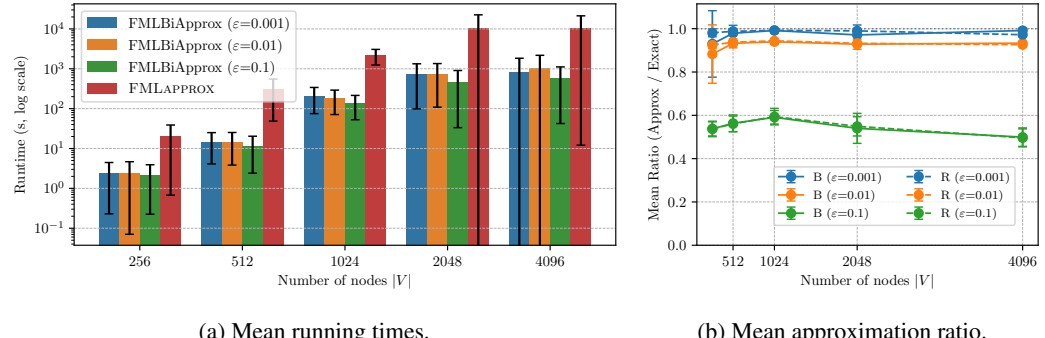

(a) Mean running times.

(b) Mean approximation ratio.

Figure 2: Comparison of FMLBIAPPROX and FMLAPPROX. The results show the mean and standard deviation as error bars over ten independent runs.

Table 2: Results on the real-world network.

| Epsilon | Runtime (s) | Accuracy (%) |
|---------|-------------|--------------|
| 0.01 | $3625.0 \pm 420.7$ | 93.1 |
| 0.001 | $4661.5 \pm 991.3$ | 99.3 |

## 6.3 Practical scalability

We sampled a connected 20,000-node induced subgraph (201,900 edges) from the Pokec dataset[1] [39]. We randomly selected one node as the terminal and colored 10% of the remaining nodes (balanced 50/50) using the binary gender attribute (Blue = male, Red = female). We then ran FMLBIAPPROX with full coverage requirement ($\alpha = 1.0$) and two precision settings ($\varepsilon = 0.01$ and $\varepsilon = 0.001$) in 10 independent runs. Table 2 shows the results. Accuracy represents the ratio of achieved coverage to the required coverage. This real-world experiment confirms the practical scalability and effectiveness of our FMLBIAPPROX algorithm, which achieves high accuracy under both precision settings.

We investigated the bicriteria guarantee's dependency on tree height $H$ on our new Pokec dataset. Over 10 runs, the average tree height was exceptionally small, $H \approx 6.68$. Our theoretical analysis provides a worst-case coverage guarantee of $(1 + \varepsilon)^{-(H+1)}$. For $\varepsilon = 0.01$, this guarantees at least $(1.01)^{-7.68} \approx 92.6\%$ coverage. Our empirically measured accuracy was 93.1%. This demonstrates that our algorithm robustly meets its formal theoretical guarantee in a real-world setting.

## 7 Limitations

**Conceptual limitations.** While the FML framework captures a fundamental trade-off between temporal resource efficiency and fairness, it assumes centralized control over the activation of communication links. This limits its applicability in decentralized or asynchronous settings such as classical federated learning [19]. Moreover, the current formulation requires predefined group membership and fixed coverage thresholds, which may be restrictive in adaptive or data-driven fairness scenarios. Finally, FML treats fairness as group-level temporal reachability, but does not model finer-grained objectives such as individual fairness or fairness over time.

**Technical limitations.** Our approximation algorithms currently focus on the case of a single terminal and two groups. While this setting already models common use cases, generalizing to arbitrary numbers of terminals and group types remains open. The exact dynamic programming algorithm has a running time complexity in $\mathcal{O}(n^5)$, limiting scalability. The bicriteria approximation improves efficiency, reducing runtime to $\mathcal{O}(n^2 + n\varepsilon^{-4} \log^4 n)$, but trades off exact fairness for scalability: its guarantees degrade with the tree height, leading to a $(1 + \varepsilon)^{\log n}$ factor in group coverage.

## 8 Conclusion and future work

We introduced the *Fair Minimum Labeling* (FML) problem, which formalizes the design of temporal activation schedules that ensure fair group-wise reachability under resource constraints. FML bridges a gap between fairness-aware graph optimization and temporal connectivity models. We proved that FML is NP-hard and $\Omega(\log n)$-hard to approximate, even under restricted settings. To address this challenge, we developed a randomized approximation framework based on probabilistic tree embeddings, achieving tight $\mathcal{O}(\log n)$ expected-cost bounds. A bicriteria variant further scales to large networks while guaranteeing bounded fairness violations. Experiments on multi-source learning tasks confirmed that FML-based methods achieve near-optimal fairness and accuracy at substantially lower cost than competing baselines. Future work includes exploiting structural properties such as low treewidth or planarity, extending the model to weighted and online settings, and designing distributed variants for decentralized or privacy-sensitive deployments. Moreover, extending FML to multiple terminals is a promising yet non-trivial direction. This would require the dynamic program to maintain terminal-reachability sets or richer state representations, thereby substantially increasing both the state space and the complexity of the merge step.

---

[1]https://snap.stanford.edu/data/soc-Pokec.html

## Acknowledgments

This work was undertaken on Barkla, part of the High Performance Computing facilities at the University of Liverpool, UK.

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

# Appendix

---

**Algorithm 2:** Bottom-up Label Computation on Tree $T$

---

**Input:** Rooted tree $T = (V_T, E_T)$ with edge weights $w(u, v)$, coloring $c : V_T \to \{B, R, \emptyset\}$.
**Output:** At each node $v \in V_T$, a set of non-dominated labels $(b, r, c)$.

**1 foreach** *node $v \in V_T$ in post-order* **do**

**2**     Initialize $\text{Labels}(v) \leftarrow \emptyset$

**3**     **if** $c(v) = B$ **then** add $(1, 0, 0)$

**4**     **if** $c(v) = R$ **then** add $(0, 1, 0)$

**5**     $\mathcal{C} \leftarrow \{(0, 0, 0)\}$

**6**     **foreach** *child $u$ of $v$* **do**

        *// Merge labels $\mathcal{C} \leftarrow \mathcal{C} \cup \text{Labels}(u)$ and keep track of edge weights*

**7**         $\mathcal{C}' \leftarrow \emptyset$

**8**         **foreach** $(b_1, r_1, c_1) \in \mathcal{C}$ **do**

**9**            Add $(b_1, r_1, c_1)$ to $\mathcal{C}'$

**10**            **foreach** $(b_2, r_2, c_2) \in \text{Labels}(u)$ **do**

**11**               Add $(b_1 + b_2, r_1 + r_2, c_1 + c_2 + w(v, u))$ to $\mathcal{C}'$

**12**         Keep only the label with minimum $c$ for each $(b, r)$ in $\text{Labels}(v)$

---

## A    Omitted Proofs

**Observation 1.** *The FML problem generalizes both the Minimum Labeling (ML) and the Minimum Steiner Labeling (MSL) problems.*

*Proof of Observation 1.* In ML, the goal is to make the entire graph temporally connected. We model this by setting the terminal set $\mathcal{T} = V$, and using a single color $c_1$ for all nodes, i.e., $\mathcal{C} = \{c_1\}$ with $c(v) = c_1$ for all $v \in V$. Additionally, we set $\rho = |\mathcal{T}|$, and we define the requirement function $f_{c_1}(V) = |V|^2$, which requires full pairwise temporal reachability among all nodes. Then, any labeling that satisfies the FML condition $r(V, V) \geq |V|^2$ must realize all-pairs temporal connectivity, which is exactly the ML objective. In MSL, the goal is to ensure temporal connectivity among a given subset of terminals $\mathcal{T} \subseteq V$. We use one color $c_1$ and assign it only to the nodes in $\mathcal{T}$ while all other nodes are not assigned any color, i.e., $\mathcal{C} = \{c_1, \emptyset\}$ and $V_{c_1} = \mathcal{T}$. Again, we set $\rho = |\mathcal{T}|$. We then define the requirement function $f_{c_1}(\mathcal{T}) = |\mathcal{T}|^2$ to require full temporal connectivity within the terminal set. The resulting FML instance enforces exactly the MSL condition, and minimizing the cost corresponds to MSL.   □

*Proof of Theorem 1.* Regarding the membership in NP, given an instance of FML with graph $G = (V, E, c)$, color set $\mathcal{C}$, requirement functions $\{f_c\}_{c \in \mathcal{C}}$, terminal set $\mathcal{T} \subseteq V$, budget $k \in \mathbb{N}$, and a candidate temporal labeling $\lambda : E \to \mathcal{P}(\mathbb{N})$, we first check whether $|\lambda| \leq k$. This is a simple summation over the labeled edge sets and takes polynomial time. Next, for each color class $c \in \mathcal{C}$, we compute the total number of temporally reachable terminal nodes from nodes in $V_c$. For each $u \in V_c$ and each $v \in \mathcal{T}$, we check if there exists a temporal path from $u$ to $v$ in the temporal graph induced by $\lambda$. This can be done using a modified BFS per source node, each with time complexity polynomial in $|V| + |\lambda|$. As we have at most $|V| \cdot |\mathcal{T}|$ such checks across all color classes, the total verification time is polynomial. For each $c \in \mathcal{C}$, we verify that $r(V_c, \mathcal{T}) \geq f_c(V_c)$. If this holds for all colors, the labeling is valid.

For hardness, we reduce from the SET COVER problem, which is NP-hard and admits no $((1 - \epsilon) \log n)$-approximation unless $\text{NP} \subseteq \text{DTIME}[n^{\mathcal{O}(\log \log n)}]$.

Let $(U, \mathcal{S}, k_{SC})$ be a Set Cover instance, where $U = \{u_1, \ldots, u_m\}$ is the universe and $\mathcal{S} = \{S_1, \ldots, S_n\}$ is a collection of subsets $S_i \subseteq U$. We construct an FML instance with one terminal and one color class as follows. The graph $G = (V, E, c)$ includes a terminal node $r$, one node $s_i$ for each set $S_i \in \mathcal{S}$, one node $t_j$ for each element $u_j \in U$, and a chain of $L$ intermediate nodes $w_{i,1}, \ldots, w_{i,L}$ for each $s_i$, which simulate a labeling cost. Edges are added between each $t_j$ and

every $s_i$ such that $u_j \in S_i$, and each $s_i$ is connected to $r$ through its respective chain. Thus, the vertex set is $V = \{r\} \cup \{t_1, \ldots, t_m\} \cup \{s_1, \ldots, s_n\} \cup \{w_{i,\ell} \mid 1 \le i \le n,\ 1 \le \ell \le L\}$, and the edge set includes $\{t_j, s_i\}$ if $u_j \in S_i$ and the path edges $\{s_i, w_{i,1}\}, \{w_{i,1}, w_{i,2}\}, \ldots, \{w_{i,L}, r\}$.

We define a single color class $c_1$ and assign it to all $t_j$ nodes, i.e., $V_{c_1} = \{t_1, \ldots, t_m\}$, and set the terminal set as $\mathcal{T} = \{r\}$ and $\rho = 1$. The requirement function for color $c_1$ is defined as $f_{c_1}(V_{c_1}) = m$, requiring that all $t_j$ nodes reach $r$ via temporal paths. The budget is set to $k = m + L \cdot k_{SC}$.

Suppose there exists a cover $\mathcal{S}' = \{S_{i_1}, \ldots, S_{i_\ell}\}$ with $\ell \le k_{SC}$ such that $\bigcup_{S \in \mathcal{S}'} S = U$. Then we can construct a temporal labeling as follows. For each chosen set $S_{i_p} \in \mathcal{S}'$, we assign labels with timestamps $2, 3, \ldots, L+2$ along its path from $s_{i_p}$ to $r$. For each element $u_j$, we select some $S_{i_p} \in \mathcal{S}'$ containing $u_j$ and assign timestamp 1 to the edge $\{t_j, s_{i_p}\}$. Every $t_j$ now has a valid temporal path to $r$ via $s_{i_p}$, and the total number of labels used is $m + L \cdot \ell \le m + L \cdot k_{SC}$, so the fairness requirement is satisfied.

Conversely, suppose there is a valid labeling $\lambda$ of cost at most $k = m + L \cdot k_{SC}$ such that every $t_j$ is temporally reachable from some $s_i$ via a path to $r$. Each of the $m$ element nodes must use at least one label on an edge to a set node $s_i$, accounting for $m$ labels. For any $s_i$ to enable reachability to $r$, the entire chain of $L$ edges must be labeled with consecutive timestamps. Therefore, the number of such activated chains is at most $k_{SC}$. The corresponding sets $\mathcal{S}'$ form a set cover of $U$ of size at most $k_{SC}$.

To show the inapproximability, we use a gap-preserving reduction from Set Cover. We choose $L = \lceil m/\delta \rceil$, where $\delta > 0$ is the gap parameter. In the YES case, the optimal FML labeling cost is at most $m + L \cdot k_{SC}$, and in the NO case, the cost must exceed $m + L \cdot (1 - \delta) \ln m \cdot k_{SC}$. Since $L$ is significantly larger than $m$, the additive term $m$ becomes negligible, and the approximation gap approaches $(1 - \delta) \log m$. Because the total number of nodes in the FML instance is $n = \Theta(n_{SC} \cdot L)$ and $L = \Theta(m)$, we have $n = \Theta(n_{SC} \cdot m)$, and hence $\log n = \Theta(\log m)$. Therefore, the logarithmic hardness gap in $m$ for SET COVER translates directly to a $\Theta(\log n)$ gap for the FML instance. $\qquad\square$

*Proof of Theorem 2.* We can represent not including subtrees using an additional label $(0, 0, 0)$ at each node. Suppose a node $v$ has children $u_1, u_2, \ldots, u_\ell$ with label sets $L_1, L_2, \ldots, L_\ell$. It holds $|L_i| \le C$ with $C = \mathcal{O}(n^2)$. We can combine $L_1$ and $L_2$ into a new set $L_{1,2}$ of size at most $C$ keeping only minimal-cost labels for each pair $(b, r)$. Then combine $L_{1,2}$ with $L_3$ to get $L_{1,2,3}$, and so forth. Each pairwise merge of two sets of size at most $C$ can be done in $\mathcal{O}(C^2)$ time by considering all pairs of labels. We sum up $(b, r)$, add the costs plus weights of the edges connecting children, and then discard dominated labels. Because we immediately discard duplicates and dominated labels, the intermediate result remains of size at most $C$. Since, each merge corresponds to one child, or equivalently to one edge, and there are $n - 1$ edges, the total costs of merging is $\mathcal{O}(n^5)$ for all nodes. Therefore, the total running time is in $\mathcal{O}(n^5)$. $\qquad\square$

Before we prove Theorem 3, we show the following result.

**Lemma 1.** *Let $\varepsilon > 0$. Let $T$ be a tree with root $t$, height $H$, and edge weights $w(u, v) \ge 1$. For any node $v$ in $T$ with height $h_v$ in its subtree, $h_v \le H$ and any exact label $\ell = (b, r, c)$ at $v$, there exists a label $\ell' = (b', r', c')$ kept in the approximate set $L'_v$ at node $v$ with*

    *(i) $c' \le c$,*

    *(ii) $b' \ge b/(1 + \varepsilon)^{h_v + 1}$ and $r' \ge r/(1 + \varepsilon)^{h_v + 1}$.*

*Proof.* The proof proceeds by induction on the height $h_v$ of node $v$. For the base case consider a leaf $v$ with $h_v = 0$. Let $b_{\text{color}}(v)$ and $r_{\text{color}}(v)$ be indicator functions returning one if $v$ is blue or red, resp., and zero otherwise. The only possible exact labels are $\ell = (b_{\text{color}}(v), r_{\text{color}}(v), 0)$, with weighted cost $c = 0$. The algorithm generates and keeps this exact label $\ell' = \ell$. It holds $c' = 0 \le c = 0$ and $b' = b$. We require $b \ge b/(1 + \varepsilon)^{0+1} = b/(1 + \varepsilon)$. This holds since $b \in \{0, 1\}$ and $1 + \varepsilon \ge 1$. Similarly, $r' \ge r/(1 + \varepsilon)$. Therefore, the claim holds for leaves.

Now, for the inductive step consider an inner node $v$ with height $h_v > 0$. Assume the claim holds for all children $u_i$ of $v$. Let $h_{u_i}$ be the height of child $u_i$, so $h_{u_i} \le h_v - 1$. Consider an exact label $\ell_v = (b_v, r_v, c_v)$ formed at node $v$ by combining exact labels $\ell_{u_i} = (b_{u_i}, r_{u_i}, c_{u_i})$ from a subset

$S \subseteq \{1, \ldots, \ell\}$ of its children, plus the node $v$'s own color. The exact counts and weighted cost are

$$b_v = \sum_{i \in S} b_{u_i} + b_{\text{color}}(v) \quad \text{and} \quad r_v = \sum_{i \in S} r_{u_i} + r_{\text{color}}(v)$$

$$c_v = \sum_{i \in S} (c_{u_i} + w(v, u_i)).$$

By the inductive hypothesis applied to each child $u_i$ (for $i \in S$), there exists a label $\ell'_{u_i} = (b'_{u_i}, r'_{u_i}, c'_{u_i})$ kept in $L'_{u_i}$ such that: $c'_{u_i} \leq c_{u_i}$, $b'_{u_i} \geq b_{u_i}/(1+\varepsilon)^{h_{u_i}+1}$, and $r'_{u_i} \geq r_{u_i}/(1+\varepsilon)^{h_{u_i}+1}$.

The approximation algorithm at node $v$ generates a candidate label $\ell''_v = (b''_v, r''_v, c''_v)$ by combining these kept labels $\ell'_{u_i}$ from the children with

$$b''_v = \sum_{i \in S} b'_{u_i} + b_{\text{color}}(v) \quad \text{and} \quad r''_v = \sum_{i \in S} r'_{u_i} + r_{\text{color}}(v)$$

$$c''_v = \sum_{i \in S} (c'_{u_i} + w(v, u_i)).$$

Now, for the costs we have $c''_v = \sum_{i \in S}(c'_{u_i} + w(v, u_i)) \leq \sum_{i \in S}(c_{u_i} + w(v, u_i)) = c_v$. Thus, $c''_v \leq c_v$. Moreover, for $b''_v$ it follows

$$b''_v = \sum_{i \in S} b'_{u_i} + b_{\text{color}}(v) \geq \sum_{i \in S} \frac{b_{u_i}}{(1+\varepsilon)^{h_{u_i}+1}} + b_{\text{color}}(v).$$

Since $h_{u_i} \leq h_v - 1$, we have $(1+\varepsilon)^{h_{u_i}+1} \leq (1+\varepsilon)^{(h_v-1)+1} = (1+\varepsilon)^{h_v}$.

$$b''_v \geq \sum_{i \in S} \frac{b_{u_i}}{(1+\varepsilon)^{h_v}} + \frac{b_{\text{color}}(v)}{(1+\varepsilon)^{h_v}} \quad (\text{since } (1+\varepsilon)^{h_v} \geq 1)$$

$$\geq \frac{1}{(1+\varepsilon)^{h_v}} \left( \sum_{i \in S} b_{u_i} + b_{\text{color}}(v) \right) = \frac{b_v}{(1+\varepsilon)^{h_v}}.$$

Similarly, we have $r''_v \geq r_v/(1+\varepsilon)^{h_v}$.

Now, this candidate label $\ell''_v = (b''_v, r''_v, c''_v)$ falls into a specific $(b, r)$-bucket defined by the geometric grid. The algorithm selects one label $\ell'_v = (b'_v, r'_v, c'_v)$ from all candidates falling into this bucket to keep in the final set $L'_v$, specifically the one with the minimum cost $c'_v$.

Since $\ell''_v$ was a candidate in this bucket with cost $c''_v$, the chosen label $\ell'_v$ must satisfy $c'_v \leq c''_v$. Combined with $c''_v \leq c_v$, we get $c'_v \leq c_v$, proving part (i) of the claim for node $v$.

Because $\ell'_v$ and $\ell''_v$ are in the same $(b, r)$-bucket defined by ratio $(1+\varepsilon)$, their counts are related by the bucket definition: $b'_v \geq b''_v/(1+\varepsilon)$ and $r'_v \geq r''_v/(1+\varepsilon)$. (If $b''$ or $r''$ is 0, the inequality holds trivially).

Combining the bounds we prove part (ii) with

$$b'_v \geq \frac{b''_v}{1+\varepsilon} \geq \frac{b_v/(1+\varepsilon)^{h_v}}{1+\varepsilon} = \frac{b_v}{(1+\varepsilon)^{h_v+1}} \quad \text{and} \quad r'_v \geq \frac{r''_v}{1+\varepsilon} \geq \frac{r_v}{(1+\varepsilon)^{h_v+1}}.$$

$\square$

We are now ready to prove Theorem 3.

*Proof of Theorem 3.* Let $\ell_{\text{opt}} = (b_{\text{opt}}, r_{\text{opt}}, c_{\text{opt}})$ be an optimal exact label at the root $t$ (node height $H$) satisfying $b_{\text{opt}} \geq \alpha|B|$ and $r_{\text{opt}} \geq \alpha|R|$. By Lemma 1 applied at the root node $t$, there exists a label $\ell' = (b', r', c')$ kept in the final approximate set $L'_t$ with $c' \leq c_{\text{opt}}$ which immediately verifies part (i). Moreover, it holds with $\xi = (1+\varepsilon)^{H+1}$ that

$$b' \geq b_{\text{opt}}/(1+\varepsilon)^{H+1} = b_{\text{opt}}/\xi \quad \text{and} \quad r' \geq r_{\text{opt}}/(1+\varepsilon)^{H+1} = r_{\text{opt}}/\xi$$

For part (ii), from $b_{\text{opt}} \geq \alpha|B|$ and $r_{\text{opt}} \geq \alpha|R|$ it follows $b' \geq (\alpha|B|)/\xi$ and $r' \geq (\alpha|R|)/\xi$. $\square$

*Proof of Theorem 4.* The cost of the final solution in the graph $G$ is the total number of temporal edge activations, i.e.,

$$|\lambda_G| \leq \sum_{(\{u,v\},t)\in\lambda_T} |P_{uv}| = \sum_{(\{u,v\},t)\in\lambda_T} d_G(u,v).$$

We can relate this cost to the optimal cost $k_G^*$ of the original FML problem in $G$ using the properties of the embedding and the weighted tree cost $C_T = \sum d_T(u,v)$ optimized by our tree algorithm.

By non-contraction ($d_G(u,v) \leq d_T(u,v)$ for all $u,v$), we have

$$|\lambda_G| \leq \sum_{(\{u,v\},t)\in\lambda_T} d_G(u,v) \leq \sum_{(\{u,v\},t)\in\lambda_T} d_T(u,v) = C_T.$$

Let $\lambda_G^*$ be an optimal solution in $G$ with cost $k_G^* = |\lambda_G^*|$. This solution induces some temporal structure on the tree $T$. Let $C_T(\lambda_G^*)$ be the corresponding weighted cost on the tree required to support the reachability of $\lambda_G^*$.

The expected stretch property implies (summing over the paths corresponding to $\lambda_G^*$):

$$\mathbb{E}_{T\sim\mathcal{D}}[C_T(\lambda_G^*)] \leq \mathcal{O}(\log n) \sum_{(\{u,v\},t)\in\lambda_G^*} d_G(u,v) = \mathcal{O}(\log n)k_G^*.$$

The optimal weighted cost on the tree, $C_{T,\text{opt}}$, must satisfy $C_{T,\text{opt}} \leq C_T(\lambda_G^*)$, as $\lambda_G^*$ provides one way to achieve the required reachability on $T$.

Our tree algorithm finds a solution $\lambda_T$ with weighted cost $C_T$ such that $C_T \leq C_{T,\text{opt}}$, either exactly, or via the approximation. Therefore, it follows $|\lambda_G| \leq C_T \leq C_{T,\text{opt}} \leq C_T(\lambda_G^*)$.

Finally, by taking the expectations over the random choice of the tree $T \sim \mathcal{D}$, we have

$$\mathbb{E}[|\lambda_G|] \leq \mathbb{E}[C_T(\lambda_G^*)] \leq \mathcal{O}(\log n)k_G^*.$$

Let $G = (V,E)$ with $|V| = n$, $|E| = m$. The distances in $G$ and the FRT embedding can be computed in $\mathcal{O}(n^2)$. Using the exact DP of Section 5.1, the overall expected running time is

$$T_{\text{exact}}(n,m) = \mathcal{O}(n^5)$$

due to the exact tree DP, see Theorem 2. The projection back to $G$ needs $\mathcal{O}(|\lambda_T| + m) \subseteq \mathcal{O}(n^2)$ time.

Using the bicriteria $(\mathcal{O}(\log n), \xi)$-approximation the expected running time is

$$T_{\text{approx}}(n,m,\varepsilon) = \mathcal{O}(n^2 + n\varepsilon^{-4}\log^4 n)$$

where the second term is due to the $n \cdot \left(\frac{1}{\varepsilon^2}\log^2 n\right)^2$ pairwise merges of label buckets. The projection step again fits in the same bound. $\qquad\square$

## B  Application Example

We give a concrete high-impact application scenario: Consider a medical AI initiative deployed in a developing region to collect patient data for training a diagnostic model. The setting involves remote rural villages with limited connectivity and urban health hubs with better infrastructure. Devices operate under tight energy constraints and are duty-cycled, i.e., radios are only active during coordinated upload windows to conserve battery.

- **Nodes $V$:** Participants' smartphones, community health hubs, and the central server.
- **Groups $V_B$, $V_R$:** For example, urban (near, well-connected) vs. rural (remote) participants; we must connect an $\alpha$-fraction of each. Alternative groupings could capture other fairness dimensions, e.g., gender or ethnic communities (demographic fairness), users with high-end vs. low-end devices (resource fairness), or solar-powered vs. grid-powered nodes (energy fairness).

- **Terminal** $t$**:** The central server.
- **Edges** $E$**:** Potential short-range (Bluetooth/Wi-Fi Direct) links to hubs and long-range links from hubs to $t$.
- **Temporal labeling** $\lambda$**:** Each activated edge is assigned a few global upload rounds, i.e., the only times radios are awake.
- **Cost:** The total number of edge-time activations (each consumes energy when radios leave sleep mode).

**Why temporality matters.** Data must traverse time-respecting paths: if a rural phone $u$ forwards to hub $v$ in round $\tau_1$, then $v$ must forward towards $t$ in a later round $\tau_2 > \tau_1$. Simply picking a static path is insufficient; we need an ordered activation plan consistent with duty-cycling.

**Outcome.** FML computes a sparse, globally scheduled activation plan with a few rounds and few active links that guarantees at least $\alpha|V_B|$ and $\alpha|V_R|$ nodes can reach $t$ temporally, while minimizing total wake-ups (edge-time labels). This aligns with real duty-cycled systems where communication is batch-scheduled to conserve energy and bandwidth.

