# OpenReview forum: "Fair Minimum Labeling: Efficient Temporal Network Activations for Reachability and Equity"
_NeurIPS.cc/2025/Conference — NeurIPS 2025 poster_

### Official Review · Reviewer_tPNL · 2025-06-28

**Clarity:** 3
**Significance:** 2
**Originality:** 4
**Rating:** 5
**Confidence:** 3

**Summary:**

This paper introduces the Fair Minimum Labeling (FML) problem, which aims to compute temporally sparse edge activation schedules in dynamic networks while ensuring fair group-wise reachability to a central terminal under global resource constraints. The FML framework formalizes the trade-off between minimizing the number of activated edges and satisfying fairness constraints across multiple groups, specifically requiring that a fixed fraction of nodes in each group temporally reach the terminal.

The authors show that the problem is NP-hard and logarithmically hard to approximate. To address this, they propose a randomized approximation framework based on probabilistic tree embeddings, leading to two algorithms: (1) an exact method on trees and (2) a more scalable bicriterial approximation that allows bounded fairness violations.

Both methods are evaluated in a multi-source learning task under resource constraints, demonstrating that FML-based solutions outperform fairness-agnostic baselines in terms of cost-efficiency and fairness. The paper also discusses practical limitations and outlines future directions, including extensions to multi-terminal and distributed settings.

**Questions:**

* The current cost model treats all edge activations equally by summing $|\lambda(e)|$ across edges. However, in real-world scenarios, activation costs may vary—e.g., long-distance links may consume more energy, low-bandwidth edges may incur higher latency, or peak-time activations may be costlier. Can the FML framework be extended to support weighted activation costs, and what challenges might arise in doing so?
* The current model enforces strictly increasing timestamps along temporal paths. In practice, however, multiple edges may be traversed simultaneously, or edge activations may involve delays (e.g., arriving at time $\tau + d$). Have you considered relaxing this assumption to allow non-strict time progression or modeling delays explicitly?
* The approximation factor of $O(\log n)$ is shown to be tight in expectation via FRT embeddings. Can you clarify whether this tightness carries over in worst-case performance as well?
* The framework uses a shared $\alpha$ for all groups. Can different coverage ratios (e.g., $\alpha_B$, $\alpha_R$) be supported? Would this affect the approximation guarantees or algorithm design?
* The fairness guarantee depends on tree height $H$. Do you have empirical data (e.g., typical values of $H$ or fairness violation) showing how this impacts coverage in practice?
* The terminal node $t$ is fixed. Since terminal placement affects reachability and cost, have you studied how the choice of $t$ (e.g., central vs. peripheral) influences fairness and performance?
* The exact DP algorithm has $O(n^5)$ time complexity, and the bicriteria method improves this to $O(n^2 + n\varepsilon^{-4} \log^4 n)$. Do you have empirical benchmarks on runtime and scalability, especially for larger graphs (e.g., $n > 10^4$)? Could sparsity or graph structure be exploited to further improve scalability?
* The current model supports only two groups and one terminal. Can it be extended to multiple groups or terminals? What new challenges would this introduce, and how would it affect scalability?
* The model assumes a static node set. In dynamic environments (e.g., nodes dropping out), how robust is FML? Can it be adapted to handle such changes without full recomputation?

**Ethical Concerns:**

["NO or VERY MINOR ethics concerns only"]

**Final Justification:**

After considering the rebuttal and discussion, I believe the authors have adequately addressed nearly all of my concerns. Below are the key points informing my final recommendation:

 * _Scalability_: The authors provided a convincing clarification on the scalability of their approach, along with additional analysis and justification.
 * _Extensions_: The authors acknowledged the limitations I pointed out and offered well-reasoned explanations for how these could be addressed or extended in future work.

**Limitations:**

Yes.

**Paper Formatting Concerns:**

The problem statement, algorithm descriptions, and empirical sections are well-organized. Figures and examples help illustrate abstract concepts effectively.

**Quality:**

3

**Strengths And Weaknesses:**

Strength:
* The paper introduces the Fair Minimum Labeling (FML) problem, which uniquely captures temporal fairness constraints in dynamic networks—a formulation not addressed in prior work.
* The authors provide formal NP-hardness and approximation lower bounds for the problem, and derive approximation algorithms, indicating tight theoretical contributions.
* The use of probabilistic tree embeddings, combined with both exact and bicriteria approximation algorithms, demonstrates a thoughtful adaptation of known techniques to a new fairness-driven setting.
* The experiments are clearly motivated and show that FML-based solutions outperform baselines in both activation cost and fairness, with controlled runtime tradeoffs.

Weakness:
* While the FML framework is motivated by real-world applications such as federated data collection, edge-cloud communication, and fair infrastructure restoration, the current formulation only supports two demographic groups and a single terminal. This limitation restricts the applicability of the proposed methods to more complex, realistic settings that involve multiple groups or multiple target nodes.
* The FML framework assumes centralized control and fixed fairness thresholds, which may not directly translate to more dynamic, decentralized environments commonly seen in federated or edge settings.
* The exact dynamic programming solution has a runtime of $\mathcal{O}(n^5)$, making it impractical for large-scale networks without further optimization. The bicriteria algorithm sacrifices exact fairness, and its violation factor grows logarithmically with the graph size, which may be undesirable in sensitive applications.
* The framework focuses solely on group-level static fairness (i.e., coverage quotas), without addressing finer-grained notions such as fairness over time or at the individual level.
* While synthetic experiments are reasonable, the evaluation would be stronger if applied to a real-world temporal network with fairness-sensitive requirements.

---

> ### Author Rebuttal · Authors · 2025-07-29
>
> We thank the reviewer for their comprehensive feedback and excellent suggestions for future work. For this paper, we focused on making a foundational contribution: introducing the Fair Minimum Labeling (FML) problem, establishing its hardness, and proposing the first algorithms with theoretical guarantees. We agree that the extensions you've suggested, such as multiple terminals and decentralized settings, are important and exciting future directions. We hope our work provides a solid platform for that research.
>
>
> **W1:** This is a fair and important limitation. For clarity and tractability, we focus on the minimal yet representative case of two groups and a single terminal, which already captures a meaningful fairness–cost trade-off and enables rigorous algorithmic design and analysis. As addressed in **Q8**, our dynamic programming formulation generalizes naturally to multiple groups by extending the state space, and the approximation guarantees remain valid. Extending to multiple terminals introduces significant algorithmic complexity, which we leave for future work and will highlight more clearly in the revised manuscript.
>
> **W2:** We agree this limits applicability in some settings, especially in fully decentralized or federated systems. However, many real-world scenarios, such as sensor networks or edge-cloud orchestration do involve centralized coordination or batch scheduling, often for energy efficiency (see also **Q1** of reviewer 7pM4). We will better emphasize these motivating examples.
>
> **W3:** The FML problem is NP-hard and log-approximation hard, so a computational trade-off is unavoidable. Our goal is not just efficiency but fairness with guarantees. While heuristic baselines are faster, they fail to ensure group-wise fairness and can yield extreme imbalance, as shown in our experiments.
> That said, our bicriteria algorithm (FMLBIAPPROX) significantly improves scalability, reducing runtime from $O(n^5)$ to near-quadratic, and handles graphs with 20k nodes efficiently (see **Q7**).
>
> **W4:** See  answer of **Q4**.
>
> **W5:** See  answer of **Q7**.
>
> **Q1:** This is an excellent question. Our framework can indeed be extended to support arbitrary, non-uniform activation costs $w(e)$, while fully preserving the $O(\log n)$ approximation guarantee. The extension is quite natural. The key is to define the metric for the FRT embedding using these costs. First, we compute a shortest-path metric $d(u,v)$ where the distance between nodes is defined as the minimum cost of a path between them, using the activation costs $w(e)$ as the edge weights. We then use FRT to embed this weighted metric into a tree T. Our DP algorithm then runs on the tree $T$, and because the tree distances being optimized are already low-distortion approximations of the minimum activation costs, the $O(\log n)$ guarantee on the final, total weighted cost holds directly. This requires no fundamental change to our algorithmic framework, demonstrating its flexibility. We will clarify this in the revised manuscript.
>
>
> **Q2:** Allowing non-strict timestamps would fundamentally change the definition of a temporal path and could introduce cycles, making reachability more complex. While our current framework does not support this, it is a well-studied variant in the temporal graph literature, and adapting our reachability definitions would be a non-trivial but interesting future direction.
>
>
>
> **Q3:** The $O(\log n)$ approximation factor holds in expectation over the random FRT embedding, which is the best known bound for tree embeddings. In the worst case for a single sampled tree, the guarantee may degrade, but the standard remedy is to sample multiple trees and select the best. Empirically, we observe low variance and high consistency across runs (see also Q5).
>
> **Q4:** Our DP formulation explicitly supports a per-group requirement function $f_c$, and therefore handles heterogeneous thresholds (e.g.,$\alpha_A \neq \alpha_B$) without modification. The approximation guarantees are unaffected as they apply pointwise per group.
>
> **Q5:** You raised an excellent point about our bicriteria algorithm's fairness guarantee, which depends on the tree height H. We investigated this on a real-world Pokec dataset (20k nodes) (see Q7). Over 10 runs, the average tree height H was exceptionally small with H≈6.68. Our theoretical analysis provides a worst-case guarantee on coverage of $(1+ε)^{-(H+1)}$. For $\varepsilon=0.01$, this guarantees a coverage of at least $(1.01)^{-7.68} ≈ 92.6\%$.
>
> Our empirically measured accuracy was 93.1%. It demonstrates that our algorithm robustly meets and even slightly exceeds its formal theoretical guarantee in a real-world setting. And, it confirms that the guarantee itself is strong, as the small tree height H prevents significant degradation. The dependency on H is not a practical weakness but a well-behaved aspect of our scalable algorithm, providing strong and reliable fairness assurances.
>
> We will include this empirical validation of the coverage bound in Section 6.
>
> **Q6:** This is an excellent point. The terminal's placement is critical and directly influences the costs associated with reaching different groups. We studied this in two ways:
>
> Our main experiment in Section 6.1 explores this by defining groups B and R based on spatial proximity to the terminal $t$ ("near" vs. "far"). The failure of the GREEDY baseline to cover the "far" group ($Coverage_R=0$) starkly illustrates how terminal placement creates cost disparities. FML is specifically designed to find low-cost solutions that overcome these structural biases.
>
> We also ran additional experiments on Barabási-Albert graphs $(n=1024)$ where we explicitly placed the terminal at either a central (high-degree) or peripheral (low-degree) node. Our bicriteria algorithm (FMLBIAPPROX) proved highly effective in both scenarios, consistently achieving over 98% of the required coverage. This demonstrates that our method is robust to the terminal's location, successfully finding fair and efficient solutions even when the terminal is not in a structurally advantageous position.
>
> We will add these observations to our empirical analysis section.
>
> **Q7:** To demonstrate practical relevance, we conducted an additional set of experiments on a real-world social network. Specifically, we sampled a connected 20,000-node induced subgraph (201,900 edges) from the Pokec dataset (available at SNAP). We randomly selected one node as the terminal and colored 10% of the remaining nodes (balanced 50/50) using the binary gender attribute (Blue = male, Red = female). We then ran FMLBIAPPROX with full coverage requirement ($\alpha = 1.0$) and two precision settings ($\varepsilon = 0.01$ and $\varepsilon = 0.001$) in ten independent runs.
>
> The results are summarized below:
>
> | Epsilon | Runtime ± Std (s) | Accuracy (%) |
> |---------|----------|-------|
> | 0.01 | 3625.0 ± 420.7 | 93.1 |
> | 0.001 | 4661.5 ± 991.3 | 99.3 |
>
> Accuracy here represents the ratio of achieved coverage to the required coverage. These real-world experiments confirm the practical scalability and effectiveness of our FMLBIAPPROX algorithm. Even on a complex social network with 20,000 nodes and nearly 202,000 edges, the algorithm achieves high accuracy under both precision settings. The results demonstrate that our method not only handles synthetic benchmark graphs efficiently but also performs reliably in realistic, large-scale settings—supporting its applicability to real-world fairness-aware network design tasks. We will include this experiment and its results in the revised manuscript.
>
> For a more systematic analysis, Figure 2a in our paper benchmarks the runtime on synthetic graphs up to $n=4096$. This plot clearly illustrates the empirical scaling and demonstrates that FMLBIAPPROX is significantly more scalable than the exact-on-tree method (FMLAPPROX), consistently outperforming it by an order of magnitude on larger instances. This confirms the practical benefit of our bicriteria algorithm, which reduces the complexity from $O(n^5)$ to a much more manageable, near-quadratic runtime in practice, as discussed in Theorem 4.
>
> Exploiting sparsity or graph structure for further scalability is also a very insightful point and a promising direction for future work. Our current framework already implicitly benefits from sparsity. The initial shortest-path metric computation on a sparse graph (where $m = O(n)$) is much faster than on a dense one.
>
> However, more explicit exploitation of structure could be possible. For instance:
>
> -   For graphs with low treewidth, specialized dynamic programming algorithms could potentially solve the problem much more efficiently, avoiding the need for tree embeddings altogether.
>
> -   The structure of planar or geometric graphs (like our synthetic data) could be used to design more efficient, specialized tree embeddings or decomposition methods.
>
>
> While these specialized approaches are beyond the scope of this paper, which focuses on providing guarantees for general graphs, they represent a key avenue for achieving even greater scalability in specific application domains.
>
>
>
> **Q8:** Extending our algorithm to $>2$ groups is conceptually straightforward: the DP state tracks coverage for each group individually, at the cost of increased dimensionality. Multiple terminals, however, introduce additional complexity. We intentionally focus on the single-terminal case to retain tractability, and clearly state this in our contribution scope.
>
>
> **Q9:** In dynamic environments (e.g., node failures or departures), partial recomputation of our labeling is feasible. The DP can be rerun locally on affected subtrees, and minor changes in the graph structure often preserve the validity of the tree embedding. While full dynamic adaptation is beyond the scope of our current work, the methods lend themselves to such extensions.
>
> We will discuss this as a natural direction for future extensions.

---

> > ### Author Response · Authors · 2025-08-05
> >
> > Thanks again for your helpful review and comments! We would appreciate it if you could let us know whether our response addressed your concerns and whether this might merit an increase in your score. We are happy to provide any further clarification if needed.

---

> > ### Comment · Reviewer_tPNL · 2025-08-05
> >
> > Thank you for your detailed responses.
> >
> > You have addressed nearly all of my concerns, including those related to _scalability_ and the _limitations_ I previously noted. I especially appreciate the thoughtful _extensions_ and _clarifications_ provided. I am satisfied with the revisions and encourage incorporating all these points into the revised version. I will raise my score accordingly.

---

> > > ### Author Response · Authors · 2025-08-05
> > >
> > > Thank you for your follow‑up and for letting us know that our responses addressed your concerns. We appreciate your positive feedback on the extensions and clarifications, and we will ensure that all these points are incorporated into the revised version.

---

### Official Review · Reviewer_68A7 · 2025-07-01

**Clarity:** 3
**Significance:** 3
**Originality:** 3
**Rating:** 4
**Confidence:** 3

**Summary:**

This paper introduces the Fair Minimum Labeling (FML) problem, addressing group-wise temporal reachability under resource constraints. It proposes a probabilistic approximation framework using tree embeddings, ensuring efficient cost and fairness for two distinct groups reaching a single terminal.

**Questions:**

- Have the authors explored more potential strategies or optimizations that could reduce the computational overhead?
- Include a more dedicated discussion on the limitations of the current assumptions and explore how they might be relaxed or adapted to more realistic environments like the mentioned distributed and federated learning.
- Double-check and ensure consistency across all quantitative results and textual interpretations.

**Ethical Concerns:**

["NO or VERY MINOR ethics concerns only"]

**Final Justification:**

The rebuttal from the authors have basically address my concerns. I have also carefully reviewed the discussions of other reviewers. From my perspective, this work is slightly above the acceptance borderline in light of the clarifications in the rebuttal and the promise to revise the manuscript from the authors.

**Limitations:**

yes

**Quality:**

2

**Strengths And Weaknesses:**

# Strengths
- The paper addresses a task with clear real-world motivation and potential application value.
- The theoretical formulation of FML and its solution is detailed and well-articulated, demonstrating a strong analytical foundation, which contribute meaningfully to the understanding of the proposed method.
# Weaknesses
- The current FML assumptions and the experiment setup appear somewhat idealized, raising questions about the general applicability and robustness of the method in practical settings.
- The proposed approach introduces a notable increase in computational time, as shown in Table 1, which could become a limiting factor for real-time or resource-constrained applications. This trade-off between performance and efficiency should be more clearly discussed.
- There seems to be a discrepancy between the results reported in Table 1 (specifically for the "approx" and "biapprox" variants) and the corresponding description in Line 344 of the manuscript.

---

> ### Author Rebuttal · Authors · 2025-07-29
>
> We thank the reviewer for their thoughtful and constructive feedback. Below, we address each of the identified weaknesses (W1–W3) and related questions (Q1–Q3).
>
> **W1/Q2:** This is an important concern, and we appreciate the opportunity to clarify.
> Our theoretical results and experiments deliberately focus on a core, minimal setting: two groups, a single terminal, and centralized scheduling. This allowed us to prove NP-hardness, establish approximation lower bounds, and design the first efficient algorithms with fairness and cost guarantees.
>
> To address the question of real-world applicability, we conducted a new large-scale experiment on a 20,000-node subgraph from the Pokec dataset (available at SNAP). We randomly selected one node as the terminal and colored 10% of the remaining nodes (balanced 50/50) using the binary gender attribute (Blue = male, Red = female). We then ran FMLBIAPPROX with full coverage requirement ($\alpha = 1.0$) and two precision settings ($\varepsilon = 0.01$ and $\varepsilon = 0.001$) in ten independent runs.
>
> The results are summarized below:
>
> | Epsilon | Runtime ± Std (s) | Accuracy (%) |
> |---------|-------------------|-------------------|
> | 0.01 | 3625.0 ± 420.7 | 93.1 |
> | 0.001 | 4661.5 ± 991.3 | 99.3 |
>
> These results show that FMLBIAPPROX maintains high coverage accuracy and scales well to large, real-world networks. We will include this experiment and its results in the revised manuscript.
>
> **Regarding the centralized model:**
> We agree that the centralized model is a limitation for some settings. However, many practical systems do fit this paradigm, e.g.:
>  - Duty-cycled sensor networks managed by a base station,
>  - Edge-cloud orchestration in bandwidth-constrained environments.
>
> Consider a medical AI initiative deployed in a developing region to collect patient data for training a diagnostic model. The setting involves remote rural villages with limited connectivity and urban health hubs with better infrastructure. Devices operate under tight energy constraints and are duty-cycled, i.e., radios are only active during coordinated upload windows to conserve battery.
>
> -   Nodes V: Participants’ smartphones, community health hubs, and the central server.
>
> -   Groups $V_B$, $V_R$​: Urban (near, well-connected) vs. rural (remote) participants; we must connect an $\alpha$-fraction of each (groups could also represent, e.g., demographic or ethnic divisions).
>
> -   Terminal $t$: The central server.
>
> -   Edges $E$: Potential short-range (Bluetooth/Wi-Fi Direct) links to hubs and long-range links from hubs to t.
>
> -   Temporal labeling $\lambda$: Each activated edge is assigned a few global upload rounds, i.e., the only times radios are awake.
>
> -   Cost: The total number of edge–time activations (each consumes energy when radios leave sleep mode).
>
> -   Why temporality matters. Data must traverse time-respecting paths: if a rural phone $u$ forwards to hub $v$ in round $\tau_1$​, then $v$ must forward towards $t$ in a later round $\tau_2>\tau_1$​. Simply picking a static path is insufficient; we need an ordered activation plan consistent with duty-cycling.
>
>
> FML computes a sparse, globally scheduled activation plan with a few rounds and few active links that guarantees at least $\alpha|V_B|$ and $\alpha|V_R|$ nodes can reach t temporally, while minimizing total wake-ups (edge–time labels). This aligns with real duty-cycled systems where communication is batch-scheduled to conserve energy and bandwidth.
>
> We will extend the discussionin the revised version.
>
>
> **W2/Q1:** This is an important and valid point. The FML problem is provably hard (NP-complete and log-approximation hard), so a computational trade-off is inevitable when seeking fairness with guarantees. Our baseline algorithms are fast precisely because they ignore fairness or cost optimization, making them inappropriate for equitable design.
>
> Nevertheless, our bicriteria algorithm FMLBIAPPROX significantly improves scalability:
> -   It reduces the runtime from $O(n^5)$ (exact DP) to $O(n^2 + n\varepsilon^{-4}\log^4 n)$;
> -   It scales to real-world graphs with 20k nodes, as demonstrated in the new Pokec experiment;
> -   More than 99.9% of the runtime is spent in the labeling phase, suggesting optimization efforts should focus there.
>
> For time-critical settings, we plan to explore in future works:
> -   Structure-aware techniques, e.g., exploiting low treewidth or planarity;
> -   Sketching and sparsification to reduce problem size;
> -   Parallelization of the DP step.
>
> We will add a discussion of these trade-offs, future acceleration strategies, and clarify the current computational bottlenecks.
>
>
> **W3/Q3:** We thank the reviewer for noting the inconsistency between Table 1 and the corresponding discussion in Line 344. Upon careful review, we identified that the labels of the FMLAPPROX and FMLBIAPPROX rows were inadvertently swapped during table formatting. The numerical results themselves are correct and consistent (i.e., FMLBIAPPROX is faster than FMLAPPROX). This has now been corrected in the revised version, and we emphasize that it does not affect any findings or interpretations in the manuscript.

---

> > ### Author Response · Authors · 2025-08-05
> >
> > Thanks again for your helpful review and comments! We would appreciate it if you could let us know whether our response addressed your concerns and whether this might merit an increase in your score. We are happy to provide any further clarification if needed.

---

> > ### Comment · Reviewer_68A7 · 2025-08-06
> >
> > I thank the author for the rebuttal. The response has basically addressed my concerns. I will increase my score and I hope these clarifications can be included in the revised version of the submisson.

---

> > > ### Author Response · Authors · 2025-08-06
> > >
> > > Thank you for your follow‑up and for letting us know that our responses addressed your concerns. We appreciate your positive feedback on the extensions and clarifications, and we will ensure that all these points are incorporated into the revised version.

---

### Official Review · Reviewer_CC3x · 2025-07-03

**Clarity:** 3
**Significance:** 2
**Originality:** 3
**Rating:** 4
**Confidence:** 3

**Summary:**

The paper proposes a new framework, termed Fair Minimum Labeling, for finding minimum-cost (temporal) paths between sources and target nodes in a graph while ensuring that the number of source nodes from different groups (types) is fair (are similar). The paper shows that the underlying optimization is NP-hard and proposes two approximate algorithms to solve the problem above. The numerical results show the method's effectiveness in lower cost and fair coverage.

**Questions:**

I would like to see an experiment on a real-world case, not just with simulation data. I think verifying the method in a more realistic dataset will strengthen the submission. I am open to increasing the score with stronger numerical results, with a real dataset (not necessarily something demanding considering the short time of the rebuttal)

**Ethical Concerns:**

["NO or VERY MINOR ethics concerns only"]

**Final Justification:**

The paper introduces an interesting problem and studies in detail a simple instance of this problem. Overall, this can be a nice addition to the conference.

**Limitations:**

Yes

**Paper Formatting Concerns:**

No.

**Quality:**

3

**Strengths And Weaknesses:**

Strengths:
- The paper is well-written, and the problem statement is clear. The problem of network optimization with fairness constraints is timely and well-motivated from a theoretical and practical perspective.
- The proofs and the math look ok. The theoretical results are precise and essential to motivate the design of approximated algorithms.
- The solution based on probabilistic tree embeddings is principled, which leverages tools from metric space for solving optimization problems on graphs.
- The authors provide code to reproduce the experiments.

Weaknesses:
- Lack of numerical experiments on real data to support the results and additional experiments to showcase the method's utility.
- Complexity of order $O(n^5)$.  Although the bicriteria approximation reduces the complexity, this comes at the cost of a dependency on the guarantees of the tree height
- The proposed solutions are limited to a single terminal and two distinct groups, making the whole introduction too complex for the final method.
- The paper assumes a central controller can decide when communication links are active. This significant assumption may not hold in decentralized or uncoordinated settings, such as classical federated learning, where device participation is often uncoordinated.
- A few citations on fairness on graphs are missing [1], [2]

[1] Navarro, Madeline, et al. "Fair GLASSO: Estimating Fair Graphical Models with Unbiased Statistical Behavior." Advances in Neural Information Processing Systems, 2024
[2] Zhou, Zhuoping, et al. "Fairness-aware estimation of graphical models." Advances in Neural Information Processing Systems, 2024

---

> ### Author Rebuttal · Authors · 2025-07-29
>
> Thank you for your thoughtful and constructive feedback. We address your points below and have added new experiments as you suggested, which we hope will merit a re-evaluation of our work.
>
> **W1/Q1:** To demonstrate practical relevance, we conducted an additional set of experiments on a real-world social network. Specifically, we sampled a connected 20,000-node induced subgraph (201,900 edges) from the Pokec dataset (available at SNAP). We randomly selected one node as the terminal and colored 10% of the remaining nodes (balanced 50/50) using the binary gender attribute (Blue = male, Red = female). We then ran FMLBIAPPROX with full coverage requirement ($\alpha = 1.0$) and two precision settings ($\varepsilon = 0.01$ and $\varepsilon = 0.001$) in ten independent runs.
>
> | Epsilon | Runtime ± Std (s) | Accuracy (%) |
> |---------|-------------------|-------------------|
> | 0.01 | 3625.0 ± 420.7 | 93.1 |
> | 0.001 | 4661.5 ± 991.3 | 99.3 |
>
>
>
>
> Accuracy here represents the ratio of achieved coverage to the required coverage. These real-world experiments confirm the practical scalability and effectiveness of our FMLBIAPPROX algorithm. Even on a complex social network with 20,000 nodes and nearly 202,000 edges, the algorithm achieves high accuracy under both precision settings. The results demonstrate that our method not only handles synthetic benchmark graphs efficiently but also performs reliably in realistic, large-scale settings—supporting its applicability to real-world fairness-aware network design tasks. We will include this experiment and its results in the revised manuscript.
>
>
>
>
> **W2:** You raised an excellent point about the bicriteria guarantee's dependency on tree height H. We investigated this on our new Pokec dataset. Over 10 runs, the average tree height was exceptionally small, H≈6.68. Our theoretical analysis provides a worst-case coverage guarantee of $(1+ε)^{-(H+1)}$. For $\varepsilon=0.01$, this guarantees at least $(1.01)^{-7.68} ≈ 92.6\%$ coverage. Our empirically measured accuracy was 93.1%. This demonstrates that our algorithm robustly meets its formal theoretical guarantee in a real-world setting and that the small H prevents significant degradation. The dependency is not a practical weakness but a well-behaved aspect of our scalable method.
>
>   We will include this empirical validation of the coverage bound in Section 6.
>
>
> **W3:** You make a fair point about the scope of our proposed solution relative to the broader problem motivated in the introduction. This was a deliberate methodological choice. Our goal with this initial work was to establish the theoretical foundations for the new and challenging FML problem. By focusing on the core, non-trivial instance of two groups and a single terminal, we were able to:
>
> -   Prove its computational hardness (NP-hard and hard to approximate).
>
> -   Design the first tractable approximation algorithms with formal guarantees.
>
>
> The introduction aims to lay out the full, practical scope of the FML problem, while our algorithms provide the first concrete and provably correct solution for a foundational version of it. We believe this approach (motivating the general problem while rigorously solving a core instance) provides a solid and necessary base for future work to build upon, such as extending the methods to the more general multi-terminal and multi-group scenarios.
>
>
>
>
> **W4:** This is a key modeling choice we made for this foundational work. The assumption of a central controller is common when first tackling complex network optimization problems, as it allows for the derivation of formal guarantees. This setting is directly applicable to many real-world scenarios, such as:
>
> -   Duty-cycled sensor networks, where a central base station computes a global activation schedule to wake up specific sensors for data collection, minimizing energy consumption across the network (see answer of **Q1** of reviewer 7pM4 for a concrete example)
>
> -   Edge-cloud systems, where a central orchestrator manages bandwidth by deciding which edge devices can push updates to the cloud at specific times.
>
>
> We acknowledge in our limitations section that this assumption does not cover fully decentralized settings like classical federated learning, and that extending our framework to such environments is an important direction for future research.
>
>
>
>
> **W5:** Thank you for pointing out the missing references. They are indeed relevant, and we will add citations and discussion of [1] and [2] to our related work section in the revised manuscript. While their objectives differ from ours, they underscore the broader interest in equitable representation in graph-based learning, reinforcing the relevance of FML.

---

> > ### Comment · Reviewer_CC3x · 2025-08-05
> >
> > I thank the reviewers for their additional input, especially regarding W1 and W2. These will improve the quality of the paper.

---

> > > ### Author Response · Authors · 2025-08-05
> > >
> > > Thank you for your follow‑up and for acknowledging our clarifications on W1 and W2. We are glad these points will help improve the quality of the paper and hope this will be reflected in your overall assessment.

---

### Official Review · Reviewer_dR4R · 2025-07-05

**Clarity:** 3
**Significance:** 3
**Originality:** 3
**Rating:** 4
**Confidence:** 2

**Summary:**

This paper introduces the Fair Minimum Labeling (FML) problem, motivated by applications like fair data collection or model updates in edge-cloud systems. The authors prove FML is NP-hard, then propose probabilistic approximation algorithms achieving tight approximation bound. Empirical evaluation on multi-source learning tasks shows that FML-based methods enforce group fairness with significantly lower activation cost than baseline heuristics while maintaining high accuracy across groups.

**Questions:**

a) Can this work be extended with more approximation/assumptions can be applied to make it more practical in runtime?

b) Can this work be used to real-world scale of network topology?

Note: I am not an expert in this area and my review focuses on the practical impact of this theoretical work. I am neutral accepting the paper, and would love more discussions from other reviewers.

**Ethical Concerns:**

["NO or VERY MINOR ethics concerns only"]

**Final Justification:**

After reading the rebuttal, i maintain my scoring of weak accept as the authors address my concerns about the practical impact of this work.

**Limitations:**

Yes

**Paper Formatting Concerns:**

N/

**Quality:**

3

**Strengths And Weaknesses:**

**Strength**

+) This paper proposes an interesting and practical problem which can be used in multiple real scenrios

+) This paper proved NP-hardness and found an algorithm to reach the theoretical bound

+) The effectiveness of the method is verified on practical problems


**Weakness**

-) Experiments are limited to synthetic random geometric graphs with two synthetic data groups. It would be better to evalute on real-world network topologies.

-) Even the approximate algorithm has runtimes quadratically which may prevent it usage in practice. It would be better to discuss if more approximation/assumptions can be applied to make it more practical?

---

> ### Author Rebuttal · Authors · 2025-07-29
>
> Thank you for your thoughtful and constructive feedback. We appreciate the opportunity to address your concerns and answer your questions.
>
> **Q (a)/W2:** Yes, and we view this as a promising direction for future work. Our primary goal in this paper is to introduce the FML problem, prove its theoretical hardness, and present the first approximation algorithms with formal fairness and cost guarantees. The proposed bicriteria method already improves scalability substantially (from $O(n^5)$ to $O(n^2 + n\varepsilon^{-4}\log^4 n))$ while preserving strong fairness guarantees in practice.
>
> Further runtime improvements could be achieved through stronger approximations or assumptions, e.g., by:
>
> -   Leveraging network structure (e.g., low treewidth, geometric constraints);
>
> -   Applying sketching methods for reducing the input size.
>
> These directions are beyond the scope of the current paper, which focuses on establishing FML as a rigorous and practically relevant framework. Nonetheless, our work lays the foundation for such future advances.
>
> **Q (b)/W1:** This is a fair point. To demonstrate the practical scalability of our approach, we conducted a new experiment on a large real-world network. Specifically, we sampled a connected 20,000-node induced subgraph (201,900 edges) from the Pokec dataset (available at SNAP). We randomly selected one node as the terminal and colored 10% of the remaining nodes (balanced 50/50) using the binary gender attribute (Blue = male, Red = female). We then ran FMLBIAPPROX with full coverage requirement ($\alpha = 1.0$) and two precision settings ($\varepsilon = 0.01$ and $\varepsilon = 0.001$) in ten independent runs.
>
> | Epsilon | Runtime ± Std (s) | Accuracy (%) |
> |---------|-------------------|-------------------|
> | 0.01 | 3625.0 ± 420.7 | 93.1 |
> | 0.001 | 4661.5 ± 991.3 | 99.3 |
>
>
>
> Accuracy here represents the ratio of achieved coverage to the required coverage. These real-world experiments confirm the practical scalability and effectiveness of our FMLBIAPPROX algorithm. Even on a complex social network with 20,000 nodes and nearly 202,000 edges, the algorithm achieves high accuracy under both precision settings. The results demonstrate that our method not only handles synthetic benchmark graphs efficiently but also performs reliably in realistic, large-scale settings—supporting its applicability to real-world fairness-aware network design tasks. We will include this experiment and its results in the revised manuscript.

---

> > ### Author Response · Authors · 2025-08-05
> >
> > Thanks again for your helpful review and comments! We would appreciate it if you could let us know whether our response addressed your concerns and whether this might merit an increase in your score. We are happy to provide any further clarification if needed.

---

### Official Review · Reviewer_7pM4 · 2025-07-05

**Clarity:** 3
**Significance:** 2
**Originality:** 2
**Rating:** 4
**Confidence:** 4

**Summary:**

This paper introduces a new problem: Fair Minimum Labeling (FML) which lies in the intersection of temporal graph algorithms and fairness in decision making. The problem asks for the minimum cost for labeling edges, each at a set of timestamps, on a given graph, such that at least a fixed portion of nodes in each color group can "temporally reach" (meaning there exists a path of edges with ) a given set of terminal nodes. The paper then gives a concrete algorithm (and its more scalable approximation version) to solving such a problem in the case of two color groups and singleton terminal node. It completes the story with proving the hardness of the problem, even for singleton node case with only one terminal. The results are complemented with experiments with practical tasks.

**Questions:**

1. I don't have much temporal graph context. I can kind of see its practical use, but it would still be nice to have a concrete example where this kind of graph is used, and show what the nodes and the temporal activation of edges embody for a real-life problem, and why the temporal information matters.
2. Are the results in Sec. 5 limited to the two-colors-plus-no-color case only? It seems the derivation of run-time uses that.
3. Do you know how the run-time scales with size of graph on the real datasets? While theory suggests polynomial run-time, in practice it could be different. How did you choose the graph size in the experiments?
4. Do you have the de-composition of your algorithms? They seem to be taking way more time than the baselines with improvements in performance. Do we know if the tree-embedding or the labeling algorithm takes more time?
5. If I understand things correctly, in the case of singleton terminal, essentially we are solving a fair connectivity problem on the static graph. While the reduction is simple and effective, the algorithm itself is not concerned with any temporal information. Is that correct? Hence we kind of lose the temporal aspect of the problem. Have you any thoughts on how to solve this problem in the multi-node terminal case?
6. If the terminal is singleton, is this problem essentially equivalent to finding a minimum cost tree that spans at least a given portion of nodes of both colors? The resulting chosen edges seem to always form a tree that connects a chosen set of nodes to the terminal node.

**Ethical Concerns:**

["NO or VERY MINOR ethics concerns only"]

**Final Justification:**

I have read the authors' rebuttal and the response of other reviewers. The authors' response has better motivated the paper with a deeper discussion on the application and provided additional experiments. I will keep my current score as I remain reserved about the theoretical complexity and the level of restrictions imposed by the assumptions made in the paper.

**Limitations:**

Yes.

**Paper Formatting Concerns:**

I didn't notice any such issues.

**Quality:**

3

**Strengths And Weaknesses:**

Strengths:
- The problem is new and seems to have practical application, so I think it is definitely worth studying. Especially, introducing fairness into problem formulation should be important for such a problem.
- The study is quite complete (at least with the given assumptions). The authors have proposed the problem, proved the hardness the found solutions that match the performance bounds.
- The algorithm seems easy to implement, so potentially usable.

Weaknesses:
- The experiments are somewhat less convincing than the theory. My major concern is that it seems too slow, especially compared to the benchmarks. Further, the self-synthesized datasets seem a bit too extreme, as it is generating two clusters of points that are in general very far away from each other. As a result GREEDY completely overlooks one of them.
- The baselines are limited, partially attributed to this problem being newly proposed. However, the chosen baselines clearly have some deficiency in either the fairness or cost perspective. GREEDY is bad for fairness because it doesn't account for it. CLOSEST is bad for cost because it blindly activates the edges along the shortest path (so it can use the same edge multiple times). Is there no simple baseline that seems to make sense for both?
- The algorithm uses some nice techniques. I'm not an expert in temporal graphs so I'm not sure how novel they are given the temporal graph contexts, but it seems to lack technical depth.

---

> ### Author Rebuttal · Authors · 2025-07-29
>
> Thank you for your thoughtful and constructive review. We address each of your points below.
>
> **W1.** We agree our methods (FMLAPPROX, FMLBIAPPROX) have higher runtime than simple heuristics. This is expected: FML is NP-hard and log-approximation hard. Our algorithms offer provable guarantees, while GREEDY and CLOSEST are fast precisely because they ignore fairness or efficiency/costs. As shown in Table 1 and Figure 2, FMLBIAPPROX achieves near-identical solution quality to the exact DP, but with significantly reduced runtime.
>
> The synthetic dataset is intentionally challenging. It simulates structural inequality (e.g., rural vs. urban access), where fairness-agnostic methods fail. This contrast highlights the need for principled algorithms like FML.
>
> We additionally added new experiments on a larger real-world data set (see answer of **Q3**).
>
>
>
> **W2.** We agree that the set of direct competitors is limited. This is a direct consequence of the novelty of the FML problem formulation, which, to our knowledge, is the first to formally combine temporal reachability, group fairness constraints, and minimum activation cost.
>
> The question of a "simple baseline that seems to make sense for both" is an excellent one. However, designing such a baseline is non-trivial and quickly approaches the complexity of the FML problem itself. Our chosen baselines represent the most natural starting points:
>
>  - GREEDY: What happens if you ignore fairness entirely and focus only on cost? (Result: unfairness).
>  - CLOSEST/ALTERNATING: What are the simplest, most direct ways to enforce fairness? (Result: high cost).
>
> They define the performance extremes, illustrating why FML’s principled approach is needed.
>
> We will add a short clarification to highlight these design choices and their limitations in Section 6.
>
> **W3.**  While our approach builds on known techniques (tree embeddings, dynamic programming), our main contribution lies in their novel integration to tackle the FML problem which is a new and challenging formulation that unifies fairness, temporal reachability, and cost minimization. We introduce:
>
>  -  The first formal definition of FML, capturing a real-world trade-off previously unaddressed;
>  -   A probabilistic tree embedding to make fairness-constrained temporal reachability tractable;
>  -   Exact and bicriteria DP algorithms that minimize activation cost while enforcing multi-group coverage;
>  -   Provable approximation guarantees that match the problem’s hardness lower bound in expectation.
>
> To our knowledge, this is the first framework to address fairness in temporal network design with rigorous algorithmic and theoretical guarantees.
>
>
>
> **Q1:** Consider a medical AI initiative deployed in a developing region to collect patient data for training a diagnostic model. The setting involves remote rural villages with limited connectivity and urban health hubs with better infrastructure. Devices operate under tight energy constraints and are duty-cycled, i.e., radios are only active during coordinated upload windows to conserve battery.
>
> -   Nodes V: Participants’ smartphones, community health hubs, and the central server.
>
> -   Groups $V_B$, $V_R$​: Urban (near, well-connected) vs. rural (remote) participants; we must connect an $\alpha$-fraction of each (groups could also represent, e.g., demographic or ethnic divisions)
>
> -   Terminal $t$: The central server.
>
> -   Edges $E$: Potential short-range radio signal (e.g., Wi-Fi Direct) links to hubs and long-range links from hubs to t.
>
> -   Temporal labeling $\lambda$: Each activated edge is assigned a few global upload rounds, i.e., the only times radios are awake.
>
> -   Cost: The total number of edge–time activations (each consumes energy when radios leave sleep mode).
>
> -   Why temporality matters. Data must traverse time-respecting paths: if a rural phone $u$ forwards to hub $v$ in round $\tau_1$​, then $v$ must forward towards $t$ in a later round $\tau_2>\tau_1$​. Simply picking a static path is insufficient; we need an ordered activation plan consistent with duty-cycling.
>
>
> FML computes a sparse, globally scheduled activation plan with a few rounds and few active links that guarantees at least $\alpha|V_B|$ and $\alpha|V_R|$ nodes can reach t temporally, while minimizing total wake-ups (edge–time labels). This aligns with real duty-cycled systems where communication is batch-scheduled to conserve energy and bandwidth.
>
>
>
> **Q2:** Yes, the algorithms and the specific complexity analysis presented in Section 5 are for the two-group (plus no-color) case. The $O(n^5)$ runtime of the exact DP, for instance, arises from merging labels of the form $(b, r, c)$, where the state space for $(b, r)$ is $O(n^2)$.
>
> However, the framework is generalizable. For $k$ distinct groups, the DP label would become $(g_1, g_2, \ldots, g_k, c)$, where $g_i$ is the count of covered nodes from group $i$. The state space for the counts would become $O(n^k)$. The pairwise merge operation would take $O((n^k)^2) = O(n^{2k})$ time. With $n$ nodes, the total runtime for the exact DP would be $O(n^{2k+1})$. This is polynomial for any fixed $k$, but the dependency on $k$ is exponential. This is a standard characteristic of such multi-parameter dynamic programming solutions. Our focus on $k=2$ provides a clear and practical starting point for this new problem area.
>
> We will briefly discuss this generalization and its complexity in the revised version.
>
>
> **Q3:** This is an excellent question about practical scalability. To address it directly, we ran our algorithm on a large, 20,000-node real-world network. We conducted an additional set of experiments on a real-world social network. Specifically, we sampled a connected 20,000-node induced subgraph (201,900 edges) from the Pokec dataset (available at SNAP). We randomly selected one node as the terminal and colored 10% of the remaining nodes (balanced 50/50) using the binary gender attribute (Blue = male, Red = female). We then ran FMLBIAPPROX with full coverage requirement ($\alpha = 1.0$) and two precision settings ($\varepsilon = 0.01$ and $\varepsilon = 0.001$) in ten independent runs.
>
> The results are summarized below:
>
> | Epsilon | Runtime ± Std (s) | Accuracy (%) |
> |---------|-------------------|-------------------|
> | 0.01 | 3625.0 ± 420.7 | 93.1 |
> | 0.001 | 4661.5 ± 991.3 | 99.3 |
>
>
>
> Accuracy here represents the ratio of achieved coverage to the required coverage. These real-world experiments confirm the practical scalability and effectiveness of our FMLBIAPPROX algorithm. Even on a complex social network with 20,000 nodes and nearly 202,000 edges, the algorithm achieves high accuracy under both precision settings. The results demonstrate that our method not only handles synthetic benchmark graphs efficiently but also performs reliably in realistic, large-scale settings, and hence supporting its applicability to real-world fairness-aware network design tasks. We will include this experiment and its results in the revised manuscript.
>
>
>
> **Q4:** The majority of the runtime, more than 99.9% across all settings, is spent on solving the dynamic program on the FRT tree. The time spent on tree embedding is negligible in comparison. This confirms that our approach already minimizes preprocessing overhead and that future runtime improvements should focus on accelerating the labeling phase.
>
>
>
> **Q5:**   While the setting simplifies with a single terminal, the temporal aspect remains essential. The solution is a temporal labeling $\lambda$ assigning timestamps to edges, and the cost is the total number of such activations. The projection step (Section 5.3) constructs time-respecting paths from the tree back to the original graph. Although the input graph is static, the constraints, objective, and solution are inherently temporal.
>
> Extending to multiple terminals is a promising but non-trivial direction. The notion of coverage would shift from "reaches $t$" to "reaches a fraction $\rho$ of terminals", requiring the DP to track which terminals are reachable or use more complex reachability structures. This would substantially increase the state space and the complexity of the merge step.
>
> We will mention multi-terminal extensions and associated challenges as a future direction.
>
> **Q6:** With a single terminal $t$, FML does boil down to “pick a minimum‑cost structure that lets at least an $\alpha$-fraction of blue and red nodes reach $t$.” In fact, the paper’s algorithms root a tree at $t$ and solve the problem on that tree before projecting the result back to the original graph.
>
> However, it is not strictly the same as a plain “minimum-cost tree spanning quotas of each color” (a quota/colored Steiner-type problem), because:
>
> -   Cost is per timestamped activation, not just per edge. An edge can be used multiple times at different timestamps and each use counts toward the cost.
> -   Feasibility depends on temporal order (strictly increasing times along each path). That constraint doesn’t exist in standard Steiner variants.
> -   After solving on an embedded tree, the solution is projected back to $G$ via shortest paths, which can introduce additional edges and is not guaranteed to stay a single static tree.
>
> -   The problem is NP-complete even in this single-terminal setting, with $\Omega(\log n)$ hardness of approximation which is stronger hardness than many classic tree problems.
>
>
>
> Your intuition that “the chosen edges seem to form a tree” is still largely right: with one sink, any cycles are wasteful, so an optimal (or near-optimal) labeling can be made cycle-free in the underlying static graph. But the temporal labeling/cost model is what keeps FML from being literally equivalent to a rooted quota Steiner tree. The paper itself notes connections to Steiner-type problems and the Balanced Connected Subgraph problem, but emphasizes the extra temporal and fairness structure.

---

> > ### Author Response · Authors · 2025-08-05
> >
> > Thanks again for your helpful review and comments! We would appreciate it if you could let us know whether our response addressed your concerns and whether this might merit an increase in your score. We are happy to provide any further clarification if needed.

---

### Note · Authors · 2025-08-12

**We thank the reviewers again for their constructive and insightful reviews and feedback, which allowed us to further refine our presentation and strengthen both the technical and empirical contributions of our work!**

Below, we summarise the main points raised and how we addressed them.

**Evaluation concerns.** In response to multiple requests (7pM4, dR4R, CC3x, 68A7, tPNL), we conducted new large-scale experiments on a 20,000-node, 201k-edge real-world social network (Pokec). Our bicriteria algorithm (FMLBIAPPROX) achieved 93–99% coverage at full fairness requirements with runtimes of about 1–1.5 h, demonstrating both scalability and robustness in realistic settings. This directly addresses prior concerns about applicability beyond synthetic graphs.

**Fairness–efficiency trade-offs.** We quantified the runtime breakdown (>99.9% in the labeling phase) and showed empirically that the bicriteria fairness guarantee remains strong in practice due to small observed tree heights (H≈6.68), matching its theoretical bound. We also corrected a minor table-labeling issue noted by R-68A7.

**Applicability.** We elaborated on how the framework:
-  Handles heterogeneous group quotas without changing guarantees.
-  Extends naturally to weighted activation costs.
 - Directly generalizes to more groups and can be extended to multiple terminals, though this introduces substantial complexity and remains a direction for future work.
 - Applies to real-world centralized scheduling scenarios (e.g., duty-cycled sensor networks, edge–cloud orchestration).

**Novelty and impact.** FML is, to our knowledge, the first formulation combining temporal reachability, group fairness constraints, and activation-cost minimization, with provable hardness and matching approximation bounds. Our algorithms are principled, practically implementable, and backed by reproducible code. The new results reinforce both the theoretical significance and practical relevance of our contribution.

We will incorporate all clarifications, empirical additions, and related-work updates, including the fairness-on-graphs citations suggested by CC3x, into the revised version. We are confident that the strengthened manuscript now makes a clear and impactful contribution.

---

### Decision · Program_Chairs · 2025-09-17

**Decision:**

Accept (poster)

**Comment:**

This paper proposes a new temporal edge activation problem in networks with fairness constraints that ensure that a proportion of each group has access to terminal nodes. This problem is shown to be hard to approximate even within a logarithmic factor. Approximation algorithms are provided for the two-group single terminal node case.
A main contribution of this paper is to introduce the fair minimum labeling problem, which is important, novel, and well-motivated. The algorithms make an interesting use of probabilistic tree embeddings and achieve strong theoretical guarantees, either exact or bi-criteria.
The restriction to the algorithms only being for two groups and a single terminal is a weakness.